# Cardiomyocyte-Specific Loss of Glutamyl-prolyl-tRNA Synthetase Leads to Disturbed Protein Homeostasis and Dilated Cardiomyopathy

**DOI:** 10.3390/cells13010035

**Published:** 2023-12-22

**Authors:** Jiangbin Wu, Jared Hollinger, Emily Bonanno, Feng Jiang, Peng Yao

**Affiliations:** 1Aab Cardiovascular Research Institute, Department of Medicine, University of Rochester School of Medicine & Dentistry, Rochester, NY 14642, USA; wyh0794@gmail.com (J.W.); jared_hollinger@urmc.rochester.edu (J.H.);; 2Undergraduate Program in Biology and Medicine, Department of Biological Sciences: Biochemistry, University of Rochester, Rochester, NY 14620, USA; ebonanno@u.rochester.edu; 3Department of Biochemistry & Biophysics, University of Rochester School of Medicine & Dentistry, Rochester, NY 14642, USA; 4The Center for RNA Biology, University of Rochester School of Medicine & Dentistry, Rochester, NY 14642, USA; 5The Center for Biomedical Informatics, University of Rochester School of Medicine & Dentistry, Rochester, NY 14642, USA

**Keywords:** cardiomyocyte, cardiomyopathy, EPRS1, heart failure, translational control

## Abstract

Glutamyl-prolyl-tRNA synthetase (EPRS1), an aminoacyl-tRNA synthetase (ARS) ligating glutamic acid and proline to their corresponding tRNAs, plays an essential role in decoding proline codons during translation elongation. The physiological function of EPRS1 in cardiomyocytes (CMs) and the potential effects of the CM-specific loss of Eprs1 remain unknown. Here, we found that heterozygous Eprs1 knockout in CMs does not cause any significant changes in CM hypertrophy induced by pressure overload, while homozygous knockout leads to dilated cardiomyopathy, heart failure, and lethality at around 1 month after Eprs1 deletion. The transcriptomic profiling of early-stage Eprs1 knockout hearts suggests a significantly decreased expression of multiple ion channel genes and an increased gene expression in proapoptotic pathways and integrated stress response. Proteomic analysis shows decreased protein expression in multi-aminoacyl-tRNA synthetase complex components, fatty acids, and branched-chain amino acid metabolic enzymes, as well as a compensatory increase in cytosolic translation machine-related proteins. Immunoblot analysis indicates that multiple proline-rich proteins were reduced at the early stage, which might contribute to the cardiac dysfunction of Eprs1 knockout mice. Taken together, this study demonstrates the physiological and molecular outcomes of loss-of-function of Eprs1 in vivo and provides valuable insights into the potential side effects on CMs, resulting from the EPRS1-targeting therapeutic approach.

## 1. Introduction

In any living organism, except viruses, translation machinery is essential for synthesizing proteins for conducting biological activities. Aminoacyl-tRNA synthetases (ARSs), a group of essential translation machinery components, are evolutionarily conserved housekeeping enzymes in bacteria, archaeal, and eukaryotes. These enzymes catalyze the ligation reaction of specific amino acids to cognate transfer RNAs (tRNAs), allowing these aminoacyl-tRNAs to be transferred to the ribosome decoding center for translating the genetic codons to amino acids for polypeptide synthesis [1,2]. Glutamyl-prolyl-tRNA synthetase (EPRS1), a cytosolic ARS that ligates glutamic acid and proline to their corresponding tRNAs, plays an essential role in decoding proline codons during translation elongation, especially for proline codon-rich pro-fibrotic extracellular matrix genes [3]. EPRS1 forms a large multi-aminoacyl-tRNA synthetase complex (MSC) with seven other enzymes and three non-ARS scaffold proteins [1]. Prior studies demonstrate the noncanonical function of EPRS1 in the transcript-selective translational silencing of pro-inflammatory mRNAs. This occurs via direct interaction with RNA structural elements in the 3′ untranslated region upon the phosphorylation-dependent release from the MSC and after interferon-gamma stimulation in monocytes and macrophages [4,5]. Biochemical analysis, using transient siRNA-mediated knockdown in Hela cells, suggests that the absence of EPRS1 leads to the degradation of multiple components of MSC, including isoleucyl-tRNA synthetase (IARS1) and two non-ARS factors, ARS complex interacting multifunctional protein 1 (AIMP1) and AIMP2 [6]. However, it is unclear whether EPRS1 is essential for the stability of MSC in vivo.

Multiple prolyl-tRNA synthetase inhibitors are currently being evaluated for treating various human diseases spanning from infectious disease to cancer [7,8,9]. Halofuginone, a Chinese medicine-derived prolyl-tRNA synthetase-specific inhibitor, and a recently developed novel compound, DWN12088, have been used in pre-clinical and clinical trials for anti-fibrosis treatment [9,10,11]. Therefore, inhibiting prolyl-tRNA synthetase activity using its specific inhibitor or a reduction in EPRS1 expression represents a promising therapeutic strategy in treating fibrosis across multiple organs [3,7,10]. Before approval of the use of any EPRS1 inhibitors for treating organ fibrosis (especially cardiac fibrosis), more evaluations in a cardiomyocyte (CM)-specific *Eprs1* knockout model are required to better understand the potential effects of EPRS1 loss-of-function as this enzyme is abundantly expressed in cardiomyocytes. We recently reported that global or myofibroblast-specific heterozygous knockout of *Eprs1* reduces proline-rich pro-fibrotic mRNA translation and antagonizes cardiac fibrosis in multiple heart failure (HF) mouse models [3]. However, the physiological function of EPRS1 in CMs and the potential effects of CM-specific loss of *Eprs1* remain unknown.

Here, we characterized the phenotypic changes in a heterozygous and homozygous cardiomyocyte specific *Eprs1* inducible knockout mouse model. Multi-omics analysis was performed to determine transcriptome-wide gene expression changes and proteomic homeostatic disturbance. Additionally, we identified proline-rich proteins that are drastically decreased at the protein expression levels upon deletion of *Eprs1*.

## 2. Materials and Methods

### 2.1. Mice

All mouse experiments were conducted in accordance with protocols approved by the University Committee on Animal Resources (UCAR) of the University of Rochester Medical Center (URMC). The mice were housed in a 12:12 h light: dark cycle in a temperature-controlled room (22 °C) in the animal housing room of URMC, with free access to water and food. *Eprs1* conditional knockout (cKO) mouse line *Eprs*_tm1c_B03 was purchased from The Center for Phenogenomics (TCP, Toronto, ON, Canada) in the form of frozen sperms and the mouse line was generated as previously described [3]. The tamoxifen-inducible transgenic mouse line *αMHC^MerCreMer^* (*αMHC^MCM/+^*) was a gift from Eric Small lab at URMC. *Eprs1* floxed mouse line *Eprs1*_tm1c_B03 (C57BL6/N-Eprs1<tm1c(EUBOMM)Hmgu>/Tcp) was purchased from The Center for Phenogenomics (TCP) in the form of frozen sperm. The *Eprs1*^flox/+^ tm1c mouse line was rederived using in vitro fertilization performed by the Mouse Genome Editing Resource at University of Rochester Medical Center. The tamoxifen-induced cardiomyocyte-specific knockout of *Eprs1* was achieved by crossing *Eprs1^flox/flox^* mice with *αMHC^MCM/+^* mice. Age- and gender-matched *αMHC^MCM/+^* mice were used as control for *Eprs1* knockout mice, and the knockout was achieved by intraperitoneal (IP) injection of tamoxifen (Sigma-Aldrich, Burlington, MA, USA, T5648-5g) at a dosage of 50 mg/kg body weight for four consecutive days. In parallel, tamoxifen was IP injected in the control mice. The knockout efficiency was confirmed using Western blot at the endpoint of each experiment. The time point of control mice undergoing euthanasia and harvest is described in figure legends.

### 2.2. Transverse Aortic Constriction Mouse Model

Age- and background-matched control (Ctrl: *αMHC^MCM/+^*) and *Eprs1* heterozygous (*Eprs1^cKO-Het^*: *Eprs1^flox/+^*; *αMHC^MCM/+^*) knockout mice at the age of ~8–12 weeks were subjected to transverse aortic constriction (TAC) surgery in this study. The tamoxifen was injected at least 3 weeks before surgery to avoid any cardiotoxicity caused by tamoxifen injection. TAC surgery was performed blindly at the Microsurgical Core of Aab Cardiovascular Research Institute (Aab CVRI) at URMC as performed previously [12]. Briefly, on the day of TAC surgery, mice were anesthetized via continuous administration of inhaled isofluorane (2%) while surgery was performed. The animals were placed supine, and a midline cervical incision was made to expose the trachea for direct intubation with 22-gauge plastic catheter. The catheter was connected to a volume-cycled ventilator supplying supportive oxygen. A right thoracotomy was performed. Stenosis was induced using a 27-gauge needle placed on the ascending aorta. Sham-operated mice underwent all aspects of the surgery besides the actual aortic ligature. A ligature was made around the needle and the aorta, completely occluding the aorta. The needle was then removed, causing severe aortic stenosis. The cardiac functions were measured using echocardiography at baseline (BL), 4 weeks, and 8 weeks post-surgery. At the endpoint of the experiment, the hearts were harvested, and sections were done by the Histology Core.

### 2.3. Echocardiography

The echocardiography was performed to measure the cardiac function of *Eprs1* heterozygous knockout mice under TAC surgery and *Eprs1* homozygous (*Eprs1^cKO-Homo^*: *Eprs1^flox/flox^*; *αMHC^MCM/+^*) knockout mice at different time points (BL, 2 weeks, 3 weeks, and 4 weeks) after tamoxifen administration. The echocardiographic image was collected using a Vevo2100 echocardiography machine (VisualSonics, Toronto, ON, Canada) and a linear-array 40 MHz transducer (MS-550D) by Surgical Core of Aab CVRI at URMC in a blinded manner. Left ventricular systolic and diastolic measurements were captured in M-mode from the parasternal short axis. Hearts were harvested at 2 weeks or 4 weeks post-tamoxifen administration for the experiments using *Eprs1* homozygous knockout mice.

### 2.4. Wheat Germ Agglutinin (WGA) Staining

The mouse hearts were fixed with 10% neutralized formalin solution and processed for paraffin-embedded sections in the Histological Core of Aab CVRI in a blinded manner. WGA staining was used to quantify the size of CMs in the murine heart. The paraffin-embedded sections were deparaffinized using the following steps: xylene (100%) for 2 × 5 min; ethanol (100%) for 2 × 5 min; ethanol (95%) for 1 × 5 min; ddH_2_O for 2 × 5 min. Antigen retrieval was performed by boiling the deparaffinized section in 10 mM citrate buffer, pH 6.0, and auto-fluorescence was quenched by incubation in 3% H_2_O_2_/PBS for 30 min at RT. At staining step, the section dots were marked by Dako pen and stained with 10 µg/mL WGA- Alex Fluor-488 (ThermoFisher, Waltham, MA, USA) for 1.5 h at RT. Then the slides were washed with PBS for 3 × 5 min and finally covered by coverslips with antifade solution (containing DAPI) for imaging. After staining with WGA-Alex Fluor-488, the cross-sectional areas were imaged by the upright fluorescence microscope (Olympus, Tokyo, Japan), and the scale bar was added to each image. The CM size was quantified using ImageJ software (v1.53). Briefly, the images were imported into the software and the scale was set based on the scale bar inside each image. A pre-designed macro, Polygon selections, was used to select the area surrounded by the WGA-positive membranes. Finally, the areas inside the WGA-positive membranes were measured as the size of cardiomyocytes.

### 2.5. Picrosirius Red Staining

Picrosirius staining was performed to measure cardiac fibrosis in the hearts using Picrosirius Red Solution (Abcam, Cambridge, UK) following the manufacturer’s instruction. Briefly, paraffin-embedded tissue sections were deparaffinized and incubated in picrosirius red solution at RT for 1 h. Then, the slides were subjected to 2 washes of 1% acetic acid and 100% ethyl alcohol, and mounted in a mounting medium. Images were captured using the PrimeHisto XE Histology Slide Scanner (Carolina) and the cardiac fibrotic area was quantified from the whole heart images of picrosirius red staining using Image J software (v1.53).

### 2.6. mRNA Expression by qRT-PCR

The RNA extraction was performed using TRIzol reagent (ThermoFisher, Waltham, MA, USA) following instructions in the manual and used for the detection of the expression of specific genes. Briefly, the tissues were homogenized in TRIzol using Minilys Personal Homogenizer (Bertin Technologies, Paris, France) and placed on ice for 15 min to lyse the tissue. The total RNA was extracted following the manual of Trizol. For the mRNA detection, 1 μg of total RNA was used as a template for reverse transcription using the iScript cDNA Synthesis Kit (Bio-Rad, Hercules, CA, USA). cDNA was diluted 5-fold and used for detecting the indicated gene expression. 18S rRNA was used as a normalizer for mRNA expression.

### 2.7. Immunoblotting

The heart homogenates were prepared in RIPA buffer (ThermoFisher Scientific, Waltham, MA, USA) containing protease inhibitor cocktail (Roche, Basel, Switzerland) and phosphatase inhibitor cocktail (Pierce, Appleton, WI, USA). The tissue lysates were centrifuged at 13,000 rpm at 4 °C for 10 min and supernatant was saved for the immunoblotting analysis. The protein concentration was measured by Bio-Rad Protein Assay dye and the lysates were incubated with SDS protein loading buffer (5X stock, National Dignostics, Atlanta, GA, USA) at 95 °C for 10 min. Finally, 20–30 μg of total protein was separated by SDS-PAGE, transferred to PVDF membranes by wet electro-transfer, blocked by 4% milk, and probed with indicated primary antibodies at 4 °C overnight. HRP-conjugated secondary antibody and ECL chemical luminescence reagents were used to detect immunoblot signal in the Gel-doc system (Bio-Rad, Hercules, CA, USA).

### 2.8. In Vivo Puromycin Incorporation Assay

For measuring protein synthesis, the control and *Eprs1* cKO mice were intraperitoneally injected with puromycin at a dose of 40 nmol/g of body weight for puromycin pulse. After 30 min post-puromycin injection, animals were sacrificed and harvested hearts were snap frozen in liquid nitrogen and stored in −80 °C for further experiments. Frozen heart samples were lysed in minilys homogenizer (Bertin technologies, Paris, France) using the RIPA lysis buffer (Pierce 89901) supplemented with 1X complete protease (Roche) and phosphatase (Pierce) inhibitor cocktail. Protein concentrations were measured with Bio-Rad Protein Assay dye and 10 μg of total protein were used for immunoblot using anti-puromycin antibody (MABE343, Sigma-Aldrich, Burlington, MA, USA). HRP-conjugated mouse secondary antibody and ECL chemiluminescence reagents were used for visualization of signals in Gel-doc system (Bio-Rad, Hercules, CA, USA).

### 2.9. Terminal Deoxynucleotidyl Transferase dUTP Nick End Labeling (TUNEL) Assay

The paraffin-embedded sections were deparaffinized through the following steps: xylene (100%) for 2 × 5 min; ethanol (100%) for 2 × 5 min; ethanol (95%) for 1 × 5 min; ddH_2_O for 2 × 5 min. The tissue sections were washed with PBS twice, permeabilized using 0.5% Triton X-100 for 5 min, and then incubated in the TUNEL reaction mixture (In Situ Cell Death Detection Kit; Sigma, 11684795910) for 1 h at 37 °C in the dark. Finally, the tissue sections were washed with PBS for 3 × 5 min, co-stained with cardiomyocyte marker α-actinin antibody, air dried, and mounted with DAPI-containing antifade medium (Vector Laboratories, Newark, CA, USA, H-1500). Images were captured using a confocal microscope (Olympus, Tokyo, Japan) and the percentage of TUNEL-positive cells was quantified using Image J software (v1.53).

### 2.10. RNA-Seq NGS Data Processing and Alignment

Total RNA extracted from hearts of Ctrl and *Eprs1^cKO-homo^* at 2 weeks post-tamoxifen injection were treated with DNase I (NEB, Ipswich, MA, USA) to remove potential genomic DNA in the RNA samples. The DNase I-treated RNA samples were purified with phenol:chloroform:isoamyl alcohol and then subjected to RNA Sequencing at the Novogene company.

RNA samples were subjected to DNA removal and polyA enrichment before library construction using NGS Library Prep. Paired-end sequencing was conducted at Novogene using NovaSeq6000 S4 with a depth of >20 M reads/sample. Reads were demultiplexed using bcl2fastq version 2.19.0. Quality filtering and adapter removal were performed using Trimmomatic version 0.36 [13] with the following parameters: “ILLUMINACLIP:2:30:10 LEADING:3 TRAILING:3 SLIDINGWINDOW:5:25 MINLEN:32 HEADCROP:10”. Processed/cleaned reads were then mapped to the Mus musculus reference genome (GRCm38) with Hisat version 2.1.1 [14] with the default settings. The subread-2.0.1 [15] package (featureCounts) was utilized to derive gene counts using the Gencode vM10 annotation [16] with the following parameters: “-T 10-g gene_name -B-C-p-ignoreDup-fracOverlap 0.1”. DESeq2 version 1.38.3 [17] was used within an R-4.2.2 [18] environment to normalize raw counts and identify significantly changed genes. Gene ontology and KEGG analysis was performed using the Enrichr or David GO webtools [19,20].

### 2.11. Quantitative Mass Spectrometry

Quantitative mass spectrometry was performed by Mass Spectrometry Resource Lab at URMC as previously described [21] to measure the protein level in the hearts of Ctrl and *Eprs1^cKO-homo^* at 2 weeks post-tamoxifen injection.

Sample Preparation: For quantitative mass spectrometry experiments, the heart lysate were run through a 4–12% SDS-PAGE gel for a short time to remove contaminants and create a ~10 mm length region, allowing the total protein to be evaluated in a single gel digest. After staining with SimplyBlue SafeStain (Invitrogen, Waltham, MA, USA), these regions were excised, cut into 1 mm cubes, de-stained, and reduced and alkylated with DTT and IAA, respectively (Sigma-Aldrich, Burlington, MA, USA). Gel pieces were dehydrated with acetonitrile. Aliquots of trypsin (Promega, Madison, WI, USA) were reconstituted to 10 ng/L in 50 mM ammonium bicarbonate and added so that the solution was just covering the dehydrated gel pieces. After 0.5 h at room temperature (RT), additional ammonium bicarbonate was added until the gel pieces were completely submerged and placed at 37 °C overnight. Peptides were extracted the next day by adding 0.1% TFA and 50% acetonitrile and dried down in a CentriVap concentrator (Labconco, Kansas City, MO, USA). Peptides were desalted with homemade C18 spin columns, dried again, and reconstituted in 0.1% TFA.

LC-MS/MS: Peptides were injected onto a homemade 30 cm C18 column with 1.8 m beads (Sepax), with an Easy nLC-1000 HPLC (ThermoFisher Scientific, Waltham, MA, USA), connected to a Q Exactive Plus mass spectrometer (ThermoFisher Scientific, Waltham, MA, USA). Solvent A was 0.1% formic acid in the water, while solvent B was 0.1% formic acid in acetonitrile. Ions were introduced to the mass spectrometer using a Nanospray Flex source operating at 2 kV. The gradient began at 3% B and held for 2 min, increased to 30% B over 41 min, increased to 70% over 3 min and held for 4 min, then returned to 3% B in 2 min and re-equilibrated for 8 min, for a total run time of 60 min. The Q Exactive Plus was operated in a data-dependent mode, with a full MS1 scan followed by 10 data-dependent MS2 scans. The full scan was conducted over a range of 400–1400 *m*/*z*, with a resolution of 70,000 at *m*/*z* of 200, an AGC target of 1 × 10^6^, and a maximum injection time of 50 ms. Ions with a charge state between 2 and 5 were picked for fragmentation. The MS2 scans were performed at a 17,500 resolution, with an AGC target of 5 × 10^4^ and a maximum injection time of 120 ms. The isolation width was 1.5 *m*/*z*, with an offset of 0.3 *m*/*z*, and a normalized collision energy of 27. After fragmentation, ions were put on an exclusion list for 15 s to allow the mass spectrometer to fragment lower abundant peptides.

Data Analysis: Raw data from MS experiments were searched using the SEQUEST search engine within the Proteome Discoverer software platform, version 2.2 (ThermoFisher Scientific, Waltham, MA, USA), and the SwissProt human database. Trypsin was selected as the enzyme allowing up to 2 missed cleavages, with an MS1 mass tolerance of 10 ppm. Samples run on the Q Exactive Plus used an MS2 mass tolerance of 25 mmu. Carbamidomethyl was set as a fixed modification, while methionine oxidation was set as a variable modification. Using the default settings, the Minora node was used to determine relative protein abundance between samples. The percolator was used as the FDR calculator, filtering out peptides with a q-value greater than 0.01.

### 2.12. Statistical Analysis

All quantitative data were presented as mean ± SEM and analyzed using Prism 8.3.0 software (GraphPad, Boston, MA, USA). The Kolmogorov–Smirnov test was used to evaluate the normal distribution of the data. For normally distributed data, unpaired two-tailed Student *t*-test was performed for the comparisons between two groups and One-way or Two-way ANOVA with Tukey’s multiple comparisons test for the comparisons between more than three groups. For not normally distributed data, the Kruskal–Wallis test with Dunn’s multiple comparisons test for the comparisons between more than three groups. The Log-rank test was performed for the comparisons of survival rate. Two-sided *p* values < 0.05 were considered to indicate statistical significance. Specific statistical methods were described in the figure legends.

## 3. Results

### 3.1. Heterozygous Knockout of Cardiomyocyte Eprs1 Does Not Affect Cardiac Function at Baseline or under Pressure Overload in Mice

To investigate the role of EPRS1 in cardiomyocytes (CMs), we generated an *αMHC*^MCM^-driven *Eprs1* conditional knockout (cKO) mouse model (*Eprs1^cKO^*) (Figure 1A). The *Eprs1* floxed mouse line *Eprs1*_tm1c_was obtained from the International Mouse Phenotyping Consortium (IMPC) [22]. The *Epr1s*^flox/+^ tm1c mouse line was bred with *αMHC*^MCM/+^ mice [23] to obtain a tamoxifen (TMX)-inducible *αMHC*-Cre-driven *Eprs1* cKO mouse line with exons 3–4 deleted from the mouse genome. Deletion of exons 3–4 resulted in an introduction of a premature termination codon TAA from exon 5 due to frame-shift (putative truncated protein sequence from *Eprs1* start codon ATG to stop codon TAA is MGAVPQRPARQAQPVVQCGALGVQTPTAFPPNSGVPVAGGAQRLCHL*). Therefore, the mRNA transcript produced from the mutated *Eprs1* gene would likely undergo nonsense-mediated mRNA decay, and the full-length EPRS1 protein could not be made. The EPRS1 protein level was reduced by ~35% in heterozygous *Eprs1^cKO^* (*Eprs1^cKO-Het^*) hearts and ~90% in homozygous *Eprs1^cKO^* (*Eprs1^cKO-Homo^*) hearts at 2 weeks after TMX injection (Figure 1B). Previous studies from GlaxoSmithKline (GSK) demonstrated the moderate inhibition of prolyl-tRNA synthetase by halofuginone protected hearts from pressure overload, neurohormonal stimulation, and acute ischemia-reperfusion injury by activating the amino acid starvation pathway [7]. To compare with the cardioprotective effect of a mild inhibition of EPRS1 activity by halofuginone, we subjected the *Eprs1^cKO-Het^* mice to transverse aortic constriction (TAC) surgery and compared the phenotypic outcomes in the control and *Eprs1^cKO-Het^* mice (Figure 1C). We observed no significant difference in left ventricular (LV) function between the control and *Eprs1^cKO-Het^* groups, as indicated by ejection fraction and fractional shortening (Figure 1D, Table 1). Moreover, no significant changes were observed in the ratio of heart weight to tibia length at the endpoint (Figure 1E). The CM size was slightly increased in *Eprs1^cKO-Het^* hearts compared with the control hearts upon TAC surgery, as indicated by wheat germ agglutinin staining, but no difference was observed at baseline (Figure 1F). Also, the fibrotic area was not changed in *Eprs1^cKO-Het^* hearts compared with the controls (Figure 1G). These results suggest that reducing around half of the EPRS1 protein expression in CMs does not cause any cardiac dysfunction or cardioprotective effects in mouse hearts at baseline and under pressure overload conditions, supporting the observation that a low-dose halofuginone-mediated inhibition of prolyl-tRNA synthetase activity does not cause severe cardiotoxicity in the pressure overload-induced HF model [7]. Moreover, the CM-specific heterozygous cKO of one allele of *Eprs* gene cannot recapitulate the cardioprotective effects from halofuginone in reducing pro-fibrotic extracellular matrix protein expression in cardiac fibroblasts [3].

### 3.2. Homozygous Knockout of Eprs1 in Cardiomyocytes Leads to Dilated Cardiomyopathy and Heart Failure in Mice

Because EPRS1 is an essential translation factor, *Eprs1^cKO-Homo^* mice underwent full penetrance of lethality in both sexes at ~28 days post-TMX-induced *Eprs1* deletion with a small number of male mice dying at ~8–10 days post-*Eprs1* deletion (Figure 2A). Weekly echocardiographic analysis suggested that *Eprs1^cKO-Homo^* mice experienced normal cardiac function for 3 weeks, followed by a dramatic decrease in ejection fraction at week 4 after TMX injections (Figure 2B,C and Table 2). This indicated a long half-life of EPRS1 protein in CMs in vivo. The hearts from *Eprs1^cKO-Homo^* mice exhibited progressive cardiac hypertrophy with chamber dilation, as shown by the significantly reduced left ventricle anterior and posterior wall diameters at diastolic and systolic phases at 2 weeks (defined as the early stage before cardiac dysfunction occurs) and 4 weeks (the late stage showing severe cardiac dysfunction) post-TMX injections (Table 2). We also observed robust cardiac fibrosis during pathological cardiac remodeling at the late stage (4 weeks post-TMX injection) (Figure 2D,E). In contrast, the cKO hearts did not show a significant difference compared with the controls in CM hypertrophy or cardiac fibrosis at the early stage (2 weeks post-TMX injection) (Figure 2D,E). Consistent with the cardiac phenotypes, the ratio of *Myh6* to *Myh7* was reduced while *Nppa* expression was increased significantly, confirming severe CM hypertrophy at the late stage (Figure 2F). Meanwhile, cardiac fibrosis marker gene expression was drastically increased, such as *Col1a1* and *Postn* (Figure 2F). Intriguingly, we observed a drastic decrease in the *Myh6*/*Myh7* ratio but no change in the expression of *Nppa*, *Col1a1*, and *Postn* in cKO hearts at the early stage, consistent with no cardiac fibrosis or heart failure at this time (Figure 2F). Terminal deoxynucleotidyl transferase dUTP nick end labeling (TUNEL) staining revealed ~1.3% CM cell death at 2 weeks post-TMX administration and ~7.3% CM cell death at 4 weeks (Figure 2G). From the corresponding areas, zoomed in on the TUNEL staining images (Appendix A), we observed that some TUNEL signals colocalized with α-actinin and some signals localized at α-actinin-deployed areas. We think the signals at α-actinin-deployed areas came from the dead CMs with degraded α-actinin and some DNA fragments remaining in these areas. The TUNEL and α-actinin co-localized signals came from CMs with early-stage apoptosis. These data suggest that the complete deletion of *Eprs1* in CMs ultimately leads to dilated cardiomyopathy and severe HF with CM death and cardiac fibrosis.

### 3.3. Early-Stage Transcriptomic Changes Indicate Cardiac Pathological Remodeling in Cardiomyocyte-Specific Eprs1 Null Hearts

To characterize the phenotypic outcome at the transcriptome-wide mRNA expression level, we conducted RNA-seq in cKO hearts 2 weeks post-*Eprs1* deletion compared with the control hearts (Figure 3A, Appendix A) before the presence of any compromised cardiac function indicated by echocardiography to avoid extensive secondary effects from the late-stage HF (Figure 2B). Principle component analysis indicated distinct overall gene expression profiles comparing control and cKO hearts (Appendix A). The correlation among biological triplicates for RNA-seq and the distribution of differentially expressed genes were evaluated (Appendix A). Among the top 20 dysregulated genes, we observed significantly reduced *Eprs1* mRNA due to the deletion of exons 3 and 4 followed by nonsense-mediated mRNA decay and *Ces1d* (carboxylesterase 1d) mRNA (Appendix A). In contrast, most of the top dysregulated genes were increased at the mRNA level, including multiple integrated stress response (ISR) marker genes (*Gdf15*, *Atf5*, *Mthfd2*, *Slc7a5*, *Aldh18a1*, and *Aars1*), pro-hypertrophy and pro-fibrosis marker genes (*Nppb*, *Map3k6*, and *Ctgf*), and a key transcription factor *Myc* required for ribosome biogenesis (Appendix A). Gene Ontology (GO) analysis revealed the activation of the ISR (GO terms including neutral amino acid transport, cellular response to amino acid starvation, tRNA aminoacylation for protein translation, cellular amino acid biosynthetic process, proline transport, etc.) and pro-apoptotic pathways (Figure 3B, left, Appendix A) as well as p53, MAPK, and mTOR signaling pathways based on KEGG (Kyoto Encyclopedia of Genes and Genomes) analysis (Figure 3C, left, Appendix A). Noticeably, there was a decrease in ion channel genes (Figure 3B, right) and the ECM-receptor interaction pathway (Figure 3C, right). Twenty ion channel genes were slightly but significantly downregulated in *Eprs1* cKO hearts, including transporters of potassium (e.g., *Kcnv2*, *Kcnj2*, *Kcnj3*, *Kcnd2*, *Kcna7*, and *Kcnh2*) and sodium ions (e.g., *Scn4b*, *Scn4a*, and *Scn10a*) (Figure 4A). Fifteen lipid metabolism-related genes (e.g., *Aox1*, *Ces1d*, *Fitm1*, etc.) were also downregulated, consistent with the metabolic switch from lipid metabolism to glycolysis during cardiac pathological remodeling [24]. Among the upregulated genes, seven p53 signaling-related genes (e.g., *Bcl2*, *Ccnd2*, *Cdkn1a*, etc.), twenty-two apoptosis-related genes (e.g, *Bcl2l2*, *Fas*, *Ddit3*, etc.), nine sarcomere genes (e.g., *Myh7*, *Lmod2*, *Myom2*, etc.), and twenty-eight transcription factors (e.g., *Myc*, *Klf2*, *Klf4*, *Klf6*, *Klf15*, *Atf3*, *Atf4*, *Atf5*, etc.) were enriched in GO analysis (Figure 4A). We confirmed the gene expression changes using RT-qPCR for the key genes with significant differential expression, including ISR (*Ddit3*, *Ddit4*) and p53 (*Cdkn1a*, *Sesn2*) pathways, transcription factor *Myc*, cardiac fibrosis marker gene *Ctgf* (upregulated genes; Figure 4B), and multiple ion channel genes (including *Kcna7*, *Kcnh2*, *Kcnv2*, etc.) (downregulated genes; Figure 4C). These results indicate a transcriptomic signature of cardiac pathological remodeling with specific compensatory responses at the mRNA expression level triggered by *Eprs1* knockout.

### 3.4. Eprs1 Loss of Function in Cardiomyocytes Triggers Proteomic Reprogramming in Mouse Hearts

EPRS1 is an essential enzyme for protein synthesis. We first utilized a puromycin incorporation assay to examine the overall changes at the protein synthesis level in control and homozygous *Eprs1* cKO hearts at the early stage (2 weeks after TMX-induced *Eprs1* cKO) to avoid massive secondary effects at the late stage. We found a mild decrease in cKO hearts compared with the controls (Appendix A). To understand the primary changes of the proteome caused by the loss of *Eprs1* at the steady state level (determined by protein synthesis and degradation rates), we performed quantitative mass spectrometry at 2 weeks post-*Eprs1* deletion (to avoid secondary effects from the late stage HF) and found a significant decrease of 312 proteins (cutoff: Log_2_FC < −1 or *p* < 0.05) and an increase of 616 proteins (cutoff: Log_2_FC > 1 or *p* < 0.05) in *Eprs1^cKO-Homo^* hearts (Figure 5A, Appendix A), indicating a significant compensatory response at this stage. GO analysis identified tRNA aminoacylation as the most highly enriched process in the downregulated proteins (Figure 5B, left). EPRS1 forms MSC with seven other ARSs and three ARS complex interacting multifunctional proteins (AIMPs) [25]. All detectable components in MSC were downregulated in *Eprs1^cKO-Homo^* hearts (Figure 5C), supporting the essential role of EPRS1 in maintaining the stability and integrity of MSC [6] in cardiomyocytes in vivo. The most severely affected MSC proteins included MARS1 (Log_2_FC = −1.63), IARS1 (Log_2_FC = −1.57), AIMP2 (Log_2_FC = −0.9), and AIMP1 (Log_2_FC = −0.8) (Figure 5C). In contrast, several ARSs outside the MSC (e.g., SARS1, AARS1, GARS1, and NARS1) were upregulated at the protein level in *Eprs1^cKO-Homo^* hearts, probably caused by the activation of the ISR as the phosphorylation of eIF2α was dramatically enhanced at both 2-week and 4-week time points in *Eprs1^cKO-Homo^* hearts (Figure 5D). We confirmed the decreased expression of MARS1, one of the ARSs in the MSC, and the increased expression of GARS1, an ARS outside the MSC (Figure 5D). Additionally, we observed a significant decrease in the eukaryotic elongation factor 1 epsilon 1 (eEF1E1) protein level (Log_2_ FC = −1.45, *p* = 7.35 × 10^−4^), supporting the direct physical interaction of eEF1E1 with EPRS1 inside the MSC [26]. In addition, fatty acid (FA) metabolism and branched-chain amino acid (BCAA) catabolic processes were also highly enriched in the downregulated proteins (Figure 5B, left; Figure 5E), suggesting that a loss of *Eprs1* preferentially reduces the protein expression of metabolic enzymes for fatty acid (e.g., ACADVL, very long-chain specific acyl-CoA dehydrogenase, and ECHS1, enoyl-CoA hydratase) and branched-chain amino acid (e.g., BCKDHB, 2-oxoisovalerate dehydrogenase subunit beta, and DBT, lipoamide acyltransferase component of branched-chain alpha-keto acid dehydrogenase complex) in CMs at the early stage (2 weeks post-TMX injection and before heart failure occurs). The decreased expression of these enzymes from pre-HF to HF stages (Figure 5F) revealed that the dysregulation of BCAA and FA metabolism may contribute partially to pathological remodeling upon the loss of *Eprs1*. Additionally, we observed reduced steady-state levels in proteins enriched in cardiac conduction and several other metabolic pathways, such as lipid, fructose, and glycogen catabolic processes (Figure 5B, left, Appendix A).

Furthermore, the upregulated proteins were enriched in translation-related processes such as the co-translational protein targeting the membrane and cytoplasmic translation (Figure 5B, right). Noticeably, a variety of ribosome proteins were significantly upregulated and enriched in the top GO terms. The mammalian target of the rapamycin complex 1 (mTORC1) pathway is known to regulate ribosome biogenesis via enhancing the translation of 5′ terminal oligopyrimidine motif-containing ribosome protein-coding mRNAs [27]. We found that the mTORC1 pathway was activated in *Eprs1^cKO-Homo^* hearts at both 2-week and 4-week time points (Figure 5G), suggesting that the activation of the mTORC1 pathway may contribute to the upregulation of ribosome proteins in *Eprs1^cKO-Homo^* hearts. As a part of the protein homeostatic reprogramming, the steady-state level was increased in proteins enriched in protein folding (e.g., HSPA2, HSPA1B, HSPB1, DNAJB5, BAX, etc.), in transport and localization (e.g., RAB5A, RAB23, LAMTOR1, ARF4, TMED5, etc.), and in autophagy (e.g., RRAGA, CAPNS1, RHEB, PRKACA, ATP6V1C1, etc.), in RNA-binding proteins involved in mRNA stabilization (e.g., RBM24, FUS, CIRBP, XRN1, KHSRP, IGF2BP2, etc.), and in muscle contraction (e.g., MYH7, MYH3, TNNC1, TNNI1, TPM3, etc.). The increased expression of these proteins may be involved in maintaining proteomic homeostasis and contractile function of CMs at the early stage of cardiac pathological remodeling.

Overlapping significantly downregulated and upregulated mRNA and proteins uncovered the consistently changed expressions of genes involved in translation and metabolism at the early stage, such as CM-specific ribosome protein RPL3L (ribosomal protein L3 like) and its homolog protein RPL3 (ribosomal protein L3) that is ubiquitously expressed (Appendix A) and phosphodiesterase (PDE1C, PDE4A) (Appendix A). Strikingly, we noticed nearly all the ribosome protein-coding mRNAs were increased with *RPL3* as the most significantly induced (Log_2_FC = 0.75, *P*_adj_ = 3.93 × 10^−19^). In contrast, the only reduced, and the second most significantly changed, ribosome protein-coding mRNA is *RPL3L* (Log_2_FC = −0.66, *P*_adj_ = 3.83 × 10^−5^), suggesting a sharp gene expression switch between these two homolog ribosome proteins under *Eprs1* loss of function. This proteomic analysis revealed disturbed protein homeostasis in *Eprs1* cKO hearts at the early stage, which might cause cardiac dysfunction at the late stage.

### 3.5. Downregulation of Proline-Rich Proteins in Cardiomyocyte-Specific Eprs1-Depleted Hearts

In our previous study, we defined the Pro-Pro dipeptidyl motif in many extracellular matrix proteins preferentially activated by EPRS1-dependent translation in cardiac fibroblasts [3]. By overlapping downregulated proteins (Log_2_ FC < −0.5) with Pro-Pro dipeptidyl motif-containing proteins, we identified 185 downregulated proteins (59.3% of 312 proteins) containing EPRS1-responsive Pro-Pro dipeptidyl motifs ^3^ and enriched in fatty acid and lipid-related metabolic enzymes based on GO analysis, such as UCP3 (mitochondrial uncoupling protein 3) and AOX1 (aldehyde oxidase 1) (Appendix A). The reason for setting a relatively flexible cut-off is to capture moderate but biologically important protein expression changes. We noticed that multiple Pro-Pro motif-containing proteins were significantly downregulated upon *Eprs1* deletion (*p* < 0.05) (Figure 6A). Using more stringent cut-offs (Log_2_ FC < −1 and *p* < 0.05), we observed the same protein hits except in UCP3 (Log_2_ FC = −0.83, *p* < 0.05) (Appendix A). We performed a Western blot to validate the decreased expression of several proteins at the early stage of cardiac remodeling (2 weeks post-TMX injection), such as TGM1 (protein-glutamine gamma-glutamyltransferase K) and UCP3 (*p* < 0.05). The other proteins also showed a clear trend of decrease, including AOX1, MYLK3 (myosin light chain kinase 3), and STK38L (serine/threonine-protein kinase 38-like), while β-actin had no change, and the total protein loading was equalized (Figure 6B and Appendix A). Taken together, our results demonstrate the indispensable function of EPRS1 in CMs for maintaining cardiac physiological function and no adverse effects of *Eprs1* heterozygous knockout in CMs on the heart. This may indicate a wide safety dose window for using EPRS1 inhibitors for treating heart failure with limited cardiotoxicity.

## 4. Discussion

Multiple missense mutations of *EPRS1* gene have been reported in humans, including compound heterozygous mutations of P14R and E205G in two patients with diabetes and bone diseases [28], the single heterozygous mutations R838H and Y791C in two sporadic cases of Parkinson’s disease [29], and five bi-allelic mutations in four patients with hypomyelinating leukodystrophy [30]. Cardiac dysfunction and heart disease were not reported in these patients, indicating that the mild or moderate loss of function of EPRS1 protein may not cause detrimental effects on the heart. In this study, our findings demonstrate that the haploinsufficiency of *EPRS1* (heterozygous cKO) in cardiomyocytes does not cause apparent cardiac disorders, while the complete depletion of *EPRS1* (homozygous cKO in CMs) leads to severe dilated cardiomyopathy, heart failure, and lethality in the mice. Halofuginone and other EPRS1-inhibitors have been tested in animals and humans for treating organ fibrosis, parasite infection, and cancer, among other diseases [10,31]. This work implies that the mild to moderate inhibition of EPRS1 by these inhibitors may not disrupt cardiomyocyte function and trigger severe cardiac toxicity (Figure 1). Given the fact that decreased EPRS1 expression or the inhibition of EPRS1 activity reduces pro-fibrotic response and extracellular matrix protein expression during fibrosis in multiple organs [3,7,32,33], and the complete loss of *EPRS1* in CMs causes severe dilated cardiomyopathy and heart failure (Figure 2), it is worthwhile evaluating further the therapeutic window for treating fibrosis using EPRS1 inhibitors to minimize the adverse effect in future.

In 2017, researchers from the GSK company showed that halofuginone protected hearts from multiple cardiac stresses by activating the amino acid starvation pathway [7]. We acknowledge that the low-dose halofuginone-mediated inhibition of prolyl-tRNA synthetase activity cannot be fully recapitulated by heterozygous *Eprs1* cKO. In the latter case, a reduced level of EPRS enzyme still maintains full catalytic activity in ligating proline to corresponding tRNA^Pro^. Also, the CM-specific heterozygous cKO of one allele of *Eprs1* gene cannot recapitulate the cardioprotective effects of halofuginone in reducing pro-fibrotic extracellular matrix protein expression in cardiac fibroblasts [3,7]. In contrast, homozygous *Eprs1* cKO leads to global translational arrest at the late stage, as indicated by the phosphorylation of eIF2α and the subsequent activation of an integrated stress response. As a result, CMs underwent apoptosis and cell death while cardiac fibroblasts were activated, leading to replacement fibrosis triggered by CM death and possibly interstitial fibrosis during cardiac pathological remodeling. Additionally, we cannot rule out the possibility of senescence induction in the cardiac remodeling process with specific compensatory responses at the mRNA level, as indicated by the increased expression of *Cdkn1a* as a key p53 pathway effector [34] at the early stage (Figure 4A,B). The mTORC1 pathway was activated at the late stage but not as much at the early stage, suggesting that *Eprs1* cKO CMs were quiescent at the early stage with p53 activation and mTOR inhibition and might become senescent at the late stage with both p53 and mTOR activation [34].

We observed a more than 4-fold EPRS1 protein decrease after two weeks post-gene knockout (Figure 5C), suggesting the half-life of the protein is ~1 week in vivo. Existing *EPRS1* mRNA and protein may maintain the function for at least 3 weeks inside the cardiomyocytes in vivo as the cKO mice did not manifest severe cardiac dysfunction until week 4 post-TMX-induced gene deletion (Figure 2C, Table 2). Alternatively, a homeostatic response may maintain cardiac function before the appearance of heart failure symptoms, as suggested by the significant increase in *Nppb*, *Nppa*, and *Myh7* at 2 weeks, as displayed in the RNA-seq (Figure 3A, Appendix A) or mass spectrometry (Figure 5A, Appendix A) analysis. Therefore, transcriptomic and proteomic analyses were performed at the early stage (2 weeks post-TMX administration) before heart failure to avoid massive changes from other cardiac cell types during cardiac dysfunction. *Eprs1* cKO causes expression changes in a specific cohort of genes at the mRNA (Figure 3 and Figure 4) and protein levels (Figure 5 and Figure 6) before any cardiac dysfunction occurs at the early stage. Among these significantly changed proteins, ARSs within the MSC were significantly reduced at the steady-state level, while the ones outside were not changed, or slightly increased (Figure 5C,D), supporting the notion that EPRS1 is essential for maintaining the integrity and stability of MSC [6] in CMs in vivo. The reduced expression of multiple MSC component proteins (RARS1, KARS1, MARS1, DARS1, AIMP1, AIMP2, QARS1, EPRS1, LARS1, IARS1) [25] may be driven by protein degradation at the co-translational levels when EPRS is absent for MSC assembly [35]. On the other hand, the loss of EPRS1 protein activates the ISR, increasing the steady-state level of ARSs outside the MSC due to induced ATF4-mediated transcriptional activation [36].

As a result of the compensatory response, ribosome proteins were dramatically increased, possibly driven by the induced expression of a ribosome biogenesis-promoting transcription factor Myc and the activation of the mTORC1 signaling pathway (Figure 4B and Figure 5G). Moreover, the overlapped genes from significantly changed mRNA and protein (RNA-seq and mass spectrometry) are minimal (Appendix A), implying uncoupled regulatory patterns of transcriptome versus translatome. However, we observed the significant upregulation of ribosome protein RPL3 and the downregulation of RPL3L, indicating that the ratio of RPL3/RPL3L is drastically increased (Appendix A). It is known that RPL3L is CM-specific and cardioprotective, while RPL3 possibly plays a detrimental role [37,38]. These integrated multi-omics data reveal multiple aspects of disrupted protein homeostasis upon genetic inactivation of *Eprs1*, including (1) a global increase in ribosome protein-coding mRNAs through increased Myc expression and mTORC1 activity [39]; (2) the inhibition of cap-dependent translation initiation by ISR activation [40]; (3) a decrease in CM-specific ribosome protein RPL3L and an increase in its homolog protein RPL3 may change ribosome heterogeneity [37,38] (more RPL3-bearing ribosome than RPL3L-bearing ribosome in the cKO hearts). Together, these events perturb the proteomic homeostasis in *Eprs1* cKO hearts, which may contribute to the progression of heart failure in mice (Figure 2).

Intriguingly, metabolic enzymes involved in fatty acid beta-oxidation and branched-chain amino acid catabolism were also significantly reduced in cKO hearts compared with the controls (Figure 5E, Appendix A). Previous studies reported a noncanonical function of phosphor-EPRS1 in activating long-chain fatty acid uptake by binding to SLC27A1 (fatty acid transport protein 1, FATP1) under hyperglycemia stress [41]. We did not observe any changes in SCL27A1 mRNA or protein expression in *Eprs1* cKO hearts, suggesting that this pathway may not be involved in the phenotype we observed in *Eprs1* cKO mice at baseline. Our previous findings showed that the halofuginone inhibition of the prolyl-tRNA synthetase activity of EPRS1 decreased the steady-state expression and polysome association of mRNAs coding for fatty acid metabolism-related proteins [3], emphasizing the canonical roles of EPRS1 in regulating fatty acid metabolism, which deserves further investigation. On the other hand, we cannot rule out the possibility of a secondary effect downstream of heart failure caused by the loss-of-function of *Eprs1* as fatty acid oxidation is often compromised and related metabolic enzyme expression is reduced in hypertensive cardiomyopathy and ischemic heart failure [24]. We also noticed that twenty ion channel mRNAs were significantly reduced in 2-week cKO hearts, including multiple potassium and sodium channels (Figure 4A). The underlying mechanism is unknown. We speculate that either one or more master transcription factors are downregulated, which reduces ion channel mRNA expression and may promote the progression of heart failure over time.

We observed multiple transcription factors coding mRNAs were significantly increased 2 weeks after *Eprs1* cKO (Figure 4A), suggesting a potential integrated transcriptional reprogramming that may not be driven by a single key transcription factor. At least 28 transcription factors (TFs) were significantly induced as a compensatory response or secondary effect, including multiple members of KLF TFs (Klf2, Klf4, Klf6), ATF TFs (Atf3, Atf4, Atf5), CEBP TFs (Cebpd, Cebpg), Myc, among others. For instance, ATF4 is a stress-responsive transcription factor induced at both transcription and translation levels by integrated stress response (in this case triggered by amino acid starvation due to EPRS1 loss of function and insufficient ligation of proline and glutamic acid to their cognate tRNAs) as indicated by the activated phosphorylation of eIF2α (Figure 5D). Therefore, ATF4 can be considered as a potential key effector TF in *Eprs1* cKO hearts as indicated by the fact that *Slc7a5*, *Atf5*, *Mthfd2*, *Gdf15*, *Aars*, and *Asns* (Figure 3A) are well-established ATF4 downstream target genes as readouts of integrated stress response pathways [36].

Prior work revealed the essential role of EPRS1 activity in embryonic development [42]. Here, our studies using a cardiomyocyte-specific *Eprs1* inducible knockout mouse model uncover the importance of the EPRS1 enzyme in maintaining cardiomyocyte and heart function at the organ and organismal levels. Future studies can further compare the phenotypic outcome and molecular changes in genetic knockout animal models of different aminoacyl-tRNA synthetases to dissect EPRS1-specific and ARS-shared effects resulting from their loss of function.

## Figures and Tables

**Figure 1 cells-13-00035-f001:**
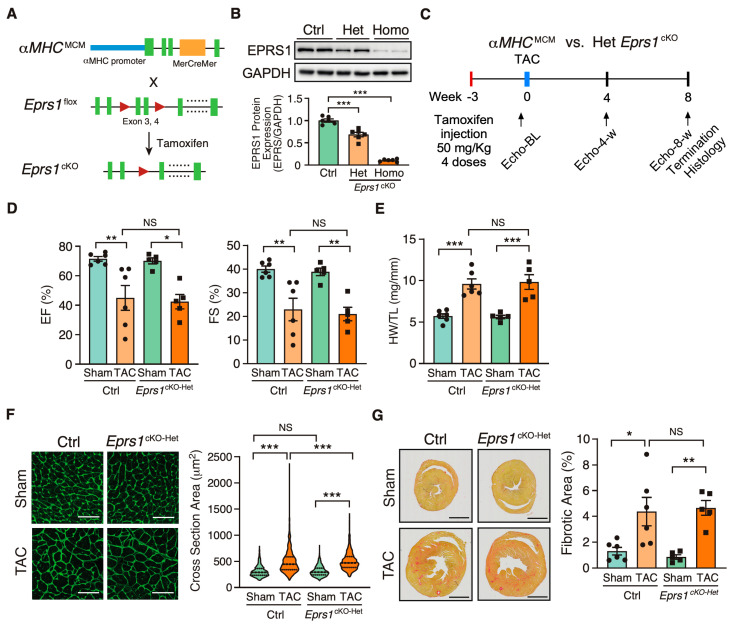
Phenotypic characterization of cardiomyocyte specific heterozygous *Eprs1* cKO mouse models under baseline and pressure overload conditions. (**A**) Strategy for generating tamoxifen-inducible CM-specific *Eprs1* conditional knockout mouse models. (**B**) EPRS1 protein expression in heterozygous (het) and homozygous (homo) *Eprs1* cKO hearts 2 weeks after tamoxifen injection. N = 6 for all three groups. (**C**) Schematic of experimental design for the control (ctrl) *aMHC*^MCM^ and heterozygous *Eprs1* cKO mice for TAC surgery and phenotyping. (**D**) Ejection fraction (EF) and fractional shortening (FS) were measured using echocardiography in control or *Eprs1^cKO-Het^* mice at 8 weeks TAC surgery or Sham operation. N = 6 for control with Sham or TAC and N = 5 for *Eprs1^cKO-Het^* with Sham or TAC. Sham-Ctrl: N = 5 M (male) + 1 F (female), Sham-Het: N = 3 M + 2 F; TAC-Ctrl: N = 4 M + 2 F; TAC-Het: N = 4 M + 1 F. (**E**) HW/TL ratio was measured in control or *Eprs1^cKO-Het^* mice at the end point of 8 weeks post-TAC surgery or Sham operation. N = 6 for Ctrl with Sham or TAC and N = 5 for *Eprs1^cKO-Het^* with Sham or TAC. (**F**) Cross-sectional area was determined through wheat germ agglutinin (WGA) staining in the sections of control or *Eprs1^cKO-Het^* hearts at the endpoint of 8 weeks post-TAC surgery or Sham operation. The quantification results are shown in the right panel. N = 4 hearts per group with >300 CMs measured per heart. Scale bar: 50 μM. (**G**) The fibrotic area was indicated by picrosirius red staining in the heart sections from control or *Eprs1^cKO-Het^* mice at 8 weeks post-TAC surgery or Sham operation. The quantification results are shown in the right panel. N = 6 for Ctrl with Sham or TAC and N = 5 for *Eprs1^cKO-Het^* with Sham or TAC. Scale bar: 2 mm. Data are represented as mean ± SEM. Comparisons were performed using One-way ANOVA with Turkey’s post hoc multiple comparisons test for (**B**), Two-way ANOVA with Turkey’s post hoc multiple comparisons test for (**D**,**E**,**G**), Kruskal–Wallis test with Dunn’s multiple comparisons test for (**F**), NS, not significant; * *p* < 0.05; ** *p* < 0.01; *** *p* < 0.001.

**Figure 2 cells-13-00035-f002:**
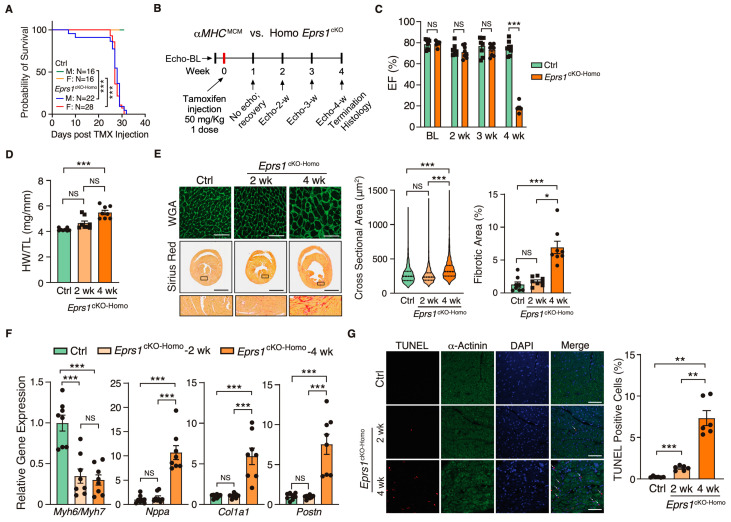
Phenotypic characterization of cardiomyocyte specific homozygous *Eprs1* cKO mouse models. (**A**) The survival rate in male and female mice after tamoxifen injection in control (Ctrl; *aMHC*^MCM^) or *Eprs1^cKO-Homo^* mice. (**B**) Schematic of experimental design for the control and homozygous *Eprs1* cKO mice for phenotyping. Control *aMHC*^MCM^ mice were harvested 4 weeks after tamoxifen injection, and homozygous *Eprs1* cKO mice were harvested 2 and 4 weeks after injection. (**C**) EF was measured weekly by echocardiography in control or *Eprs1^cKO-Homo^* mice post-tamoxifen injection. N = 8 for control and *Eprs1^cKO-Homo^*. Note: N = 7 for *Eprs1^cKO-Homo^* group at the 4-week time point because one mouse died before the echocardiography measurement. Ctrl: N = 3M + 5F; Homo: N = 7M + 1F. (**D**) HW/TL ratio was measured in control or *Eprs1^cKO-Homo^* mice at 2 weeks or 4 weeks post-tamoxifen injection. N = 8 for each group. (**E**) WGA and picrosirius red staining were performed to measure the cross-sectional area (CSA) of CMs and fibrotic area, respectively. The quantification of CSA was shown in the middle panel and the fibrotic area in the right panel. N = 4 hearts per group with >330 CMs measured per heart for CSA quantification. N = 8 for each group for fibrotic area quantification. Scale bar: 50 μM for WGA staining and 2 mm for picrosirius red staining. (**F**) The mRNA expression of hypertrophic and fibrotic marker genes in control or *Eprs1^cKO-Homo^* hearts at 2-week and 4-week time points. N = 8 for each group. (**G**) TUNEL assay co-stained with α-actinin was performed to evaluate the cell death of cardiomyocytes in the hearts of control or *Eprs1^cKO-Homo^* mice. N = 6 for each group. Scale bar: 40 μM. Data are represented as mean ± SEM. One-way ANOVA with Turkey’s post hoc multiple comparisons test for (**D**), (**E**) (right panel), (**F**,**G**), Kruskal–Wallis test with Dunn’s multiple comparisons test for (**E**) (middle panel), and Two-way ANOVA with Turkey’s post hoc multiple comparisons test for (**C**), Log-rank test for (**A**). NS, not significant; * *p* < 0.05; ** *p* < 0.01; *** *p* < 0.001.

**Figure 3 cells-13-00035-f003:**
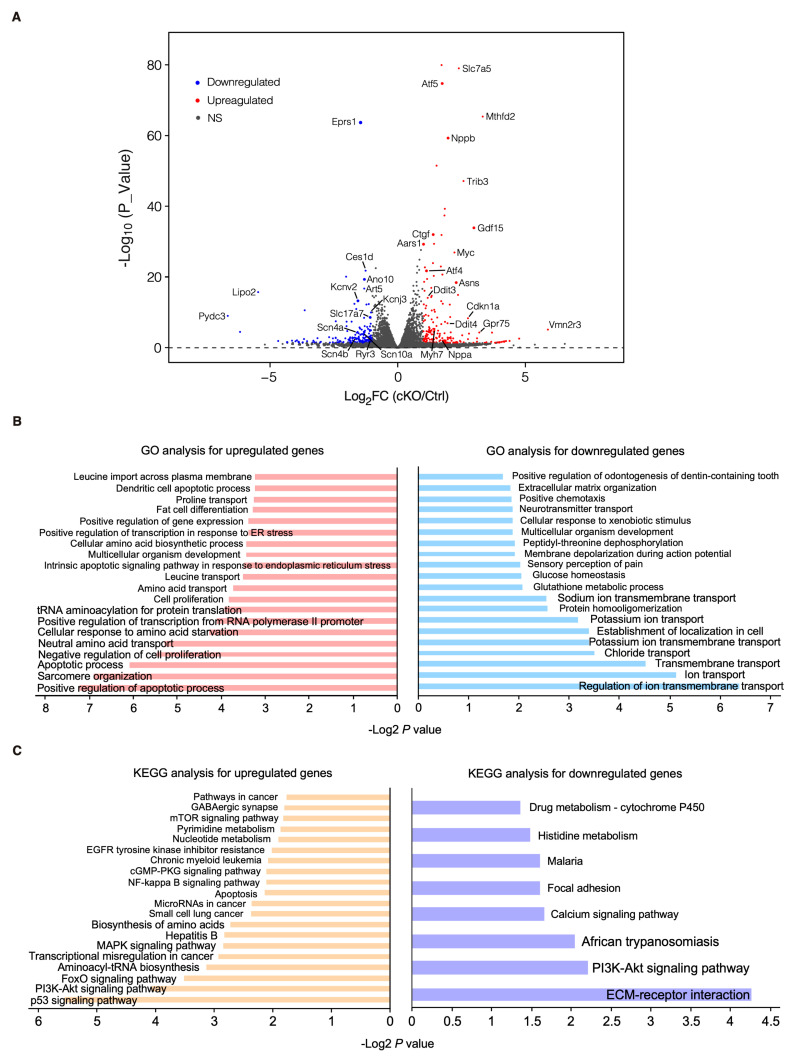
Transcriptomic profiling of control and cKO hearts at the early stage. (**A**) Volcano plot of RNA-seq in control (*αMHC*^MCM^) and homozygous *Eprs1* cKO hearts from mice at 2 weeks post-tamoxifen injection. (**B**) Gene ontology analysis (biological process) of differentially expressed genes. (**C**) KEGG pathway enrichment of differentially expressed genes.

**Figure 4 cells-13-00035-f004:**
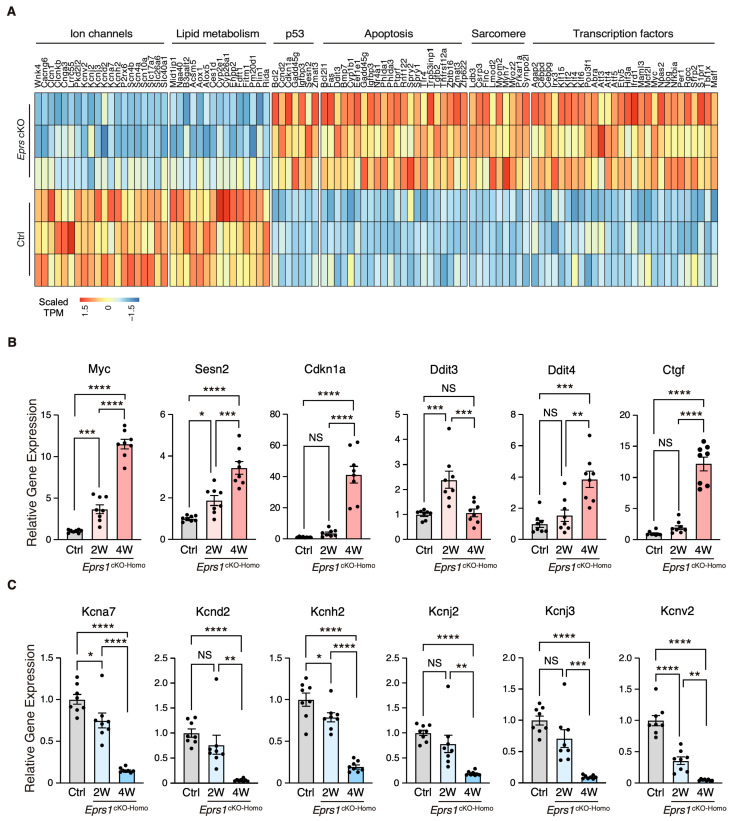
Gene expression profile of enriched gene pathways in *Eprs1* cKO hearts. (**A**) Heat map showing gene expression in enriched pathways. (**B**) RT-qPCR quantification of a selected cohort of upregulated genes. Ctrl: *aMHC*^MCM^ mice at 2 weeks post-tamoxifen injection. (**C**) RT-qPCR quantification of a selected cohort of downregulated genes. Ctrl: *aMHC*^MCM^ mice at 2 weeks post-tamoxifen injection. Data are represented as mean ± SEM. Comparisons were performed by One-way ANOVA with Turkey’s post hoc multiple comparisons test for (**B**,**C**). NS, not significant; * *p* < 0.05; ** *p* < 0.01; *** *p* < 0.001; **** *p* < 0.0001.

**Figure 5 cells-13-00035-f005:**
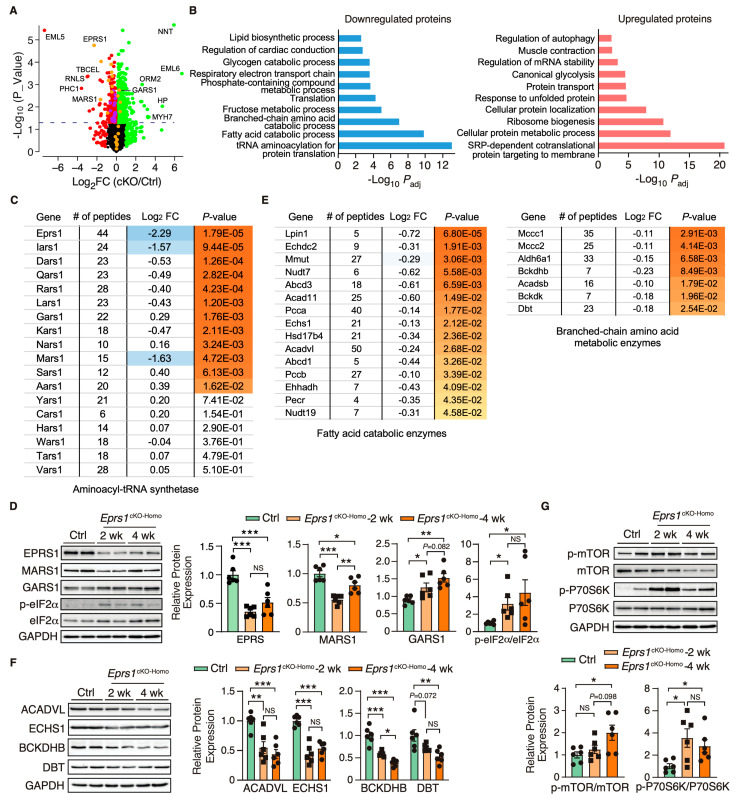
Proteomic profiling of control and *Eprs1* cKO hearts at the early stage. (**A**) The dot plot shows mass spectrometry analysis of proteomic changes at the protein steady-state level in control (*aMHC*^MCM^) and *Eprs1* cKO hearts at 2 weeks post-tamoxifen injection. The cutoff was set as |Log_2_FC (cKO/Ctrl)| > 1 or *p* < 0.05, and red dots indicate downregulated proteins and green dots for upregulated proteins (yellow dots for aminoacyl-tRNA synthetases). Purple dots represent significantly downregulated proteins encoding lipid and branched-chain amino acid metabolism. (**B**) Gene ontology analysis of downregulated protein and upregulated proteins for biological process enrichment. Top GO terms are included, and redundant pathways are removed. (**C**) Protein expression changes of cytosolic aminoacyl-tRNA synthetases as indicated by mass spectrometry analysis. (**D**) Western blot confirmed the upregulation of tRNA aminoacylation synthetases and activation of ISR in the *Eprs1* cKO hearts. Ctrl: *aMHC*^MCM^ 4 weeks post-tamoxifen injection. The quantification data are shown on the right panel. (**E**) Protein expression changes in mitochondrial fatty acid and branched-chain amino acid-related metabolic enzymes as indicated by mass spectrometry analysis. (**F**) Western blot confirmed the downregulation of BCAA metabolic enzyme and fatty acid beta-oxidation enzymes in the *Eprs1* cKO hearts. Ctrl: *aMHC*^MCM^ 4 weeks post-tamoxifen injection. The quantification data are shown on the right panel. (**G**) mTOR signaling pathway was activated in the *Eprs1* cKO hearts. Ctrl: *aMHC*^MCM^ 4 weeks post-tamoxifen injection. The quantification of p-mTOR/mTOR and p-P70S6K/P70S6K ratios are shown as bar graphs. Data are represented as mean ± SEM. Comparisons were performed by One-way ANOVA with Turkey’s post hoc multiple comparisons test for (**D**,**F**,**G**). NS, not significant; * *p* < 0.05; ** *p* < 0.01; *** *p* < 0.001.

**Figure 6 cells-13-00035-f006:**
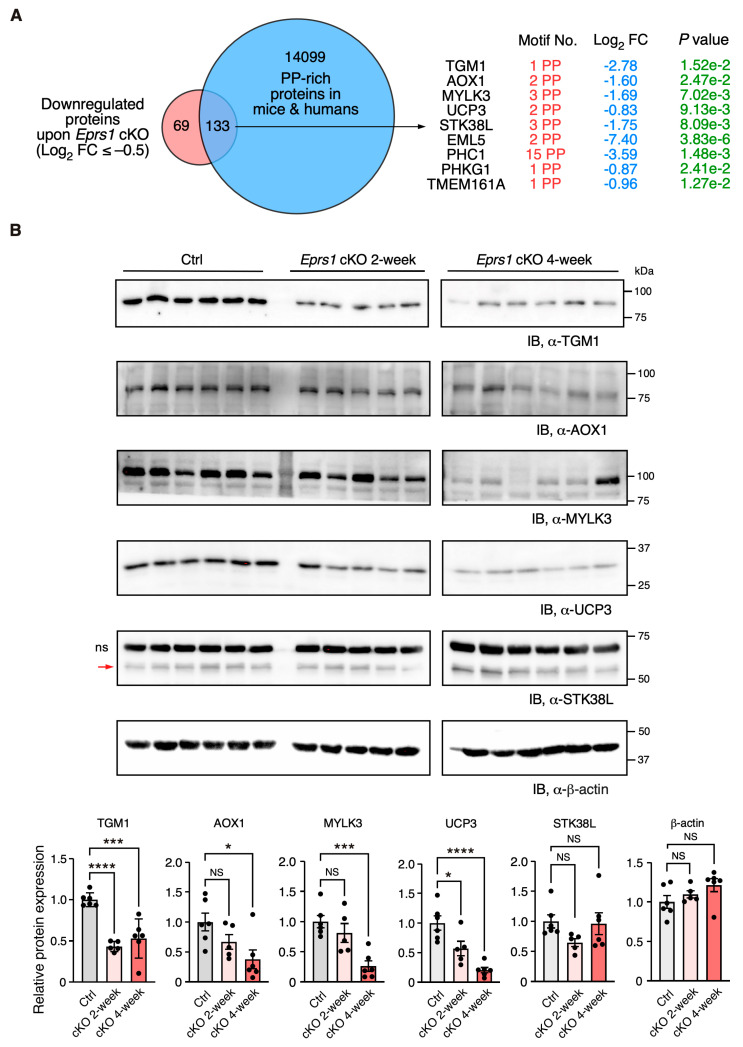
Validation of downregulation of Pro-Pro dipeptidyl motif-containing proteins in *Eprs1* cKO hearts. (**A**) Venn diagram showing downregulated proteins containing Pro-Pro dipeptidyl motifs in *Eprs1* cKO hearts. (**B**) Western blot measurement of steady-state protein levels of a selection of downregulated Pro-Pro dipeptidyl motif-bearing proteins. Heart samples from 2- and 4-weeks post-tamoxifen injection and control mice (*aMHC*^MCM^ 4 weeks post-tamoxifen injection) were used for comparisons. b-actin was used as the normalizer for quantification. ns: non-specific. STK38L is 54 KDa, and the upper non-specific band is nearly 70 KDa. Data are represented as mean ± SEM. Comparisons were performed by One-way ANOVA with Turkey’s post hoc multiple comparisons test for (**B**). NS, not significant; * *p* < 0.05; *** *p* < 0.001; **** *p* < 0.0001.

**Table 1 cells-13-00035-t001:** Echocardiographic analysis (M-mode short axis) of control (*αMHC*^MCM^) and heterozygous CM-specific *Eprs1* cKO mice up to 8 weeks post-TAC surgery compared with Sham surgery.

Parameter	Sham		TAC	
	Ctrl (N = 6)	*Eprs* cKO-Het (N = 5)	Ctrl (N = 6)	*Eprs* cKO-Het (N = 5)
Heart Rate (bpm)	616.86 ± 27.79	626.34 ± 21.24	598.83 ± 8.32	566.62 ± 21.16
Internal Diameter, Systole (mm)	2.1 ± 0.04	2.14 ± 0.11	3.23 ± 0.5	3.56 ± 0.32 ^#^
Internal Diameter, Diastole (mm)	3.5 ± 0.07	3.5 ± 0.18	4.09 ± 0.38	4.49 ± 0.29
Volume; Systole (mL)	14.5 ± 0.65	15.43 ± 1.84	50.05 ± 19.31	55.85 ± 13.07
Volume; Diastole (mL)	51.1 ± 2.63	51.92 ± 6.25	78.64 ± 18.88 ***	94.16 ± 14.92 ^##;$$^
Stroke Volume (mL)	36.6 ± 2.43	36.49 ± 4.76	28.59 ± 2.11	38.31 ± 4.7
Ejection Fraction (%)	71.4 ± 1.54	70.06 ± 2.01	44.91 ± 8.41 **	42.34 ± 4.91 ^#^
Fractional Shortening (%)	39.97 ± 1.32	38.81 ± 1.6	22.88 ± 4.78 **	20.95 ± 2.85 ^##^
Cardiac Output (mL/min)	22.57 ± 1.67	22.69 ± 2.83	17.07 ± 1.13	21.73 ± 2.88
LV Mass (mg)	76.84 ± 4.22	72.27 ± 3.44	149.5 ± 19.16 **	181.12 ± 16.09 ^###^
LV Mass Cor (mg)	61.47 ± 3.38	57.81 ± 2.76	119.6 ± 15.33 **	144.9 ± 12.87 ^###^
LV Anterior Wall Diameter, Systole (mm)	1.05 ± 0.04	1.01 ± 0.07	1.26 ± 0.06 *	1.27 ± 0.04 ^#^
LV Anterior Wall Diameter, Diastole (mm)	0.73 ± 0.05	0.73 ± 0.05	1.03 ± 0.05 **	1.04 ± 0.05 ^##^
LV Posterior Wall Diameter, Systole (mm)	1.07 ± 0.05	1.02 ± 0.04	0.98 ± 0.09	1.06 ± 0.07
LV Posterior Wall Diameter, Diastole (mm)	0.65 ± 0.02	0.6 ± 0.05	0.82 ± 0.06	0.87 ± 0.05 ^##^

Comparisons were performed using Two-way ANOVA with Turkey’s pos-hoc multiple comparisons test. *: comparing Sham vs TAC for Ctrl group. ^#^: comparing Sham vs TAC for *Eprs* cKO-Het group. ^$^: comparing Ctrl TAC and *Eprs1* cKO-Het. *, ^#^: *p* < 0.05; **, ^##^, ^$$^: *p* < 0.01; ***, ^###^: *p* < 0.001.

**Table 2 cells-13-00035-t002:** Echocardiographic analysis (M-mode short axis) of control (*αMHC*^MCM^) and homozygous CM-specific *Eprs1* cKO mice up to 4 weeks at baseline.

Parameter	Ctrl (N = 8)	*Eprs* cKO -Homo (N = 8)	Ctrl (N = 8)	*Eprs* cKO -Homo (N = 8)	Ctrl (N = 8)	*Eprs* cKO -Homo (N = 8)	Ctrl (N = 8)	*Eprs* cKO -Homo (N = 7)
	BL		2 Week		3 Week		4 Week	
Heart Rate (bpm)	542.9 ± 7.97	572.51 ± 9.94 *	562.41 ± 9.84	590.92 ± 14.85	564.01 ± 18.21	578.91 ± 7.46	559.94 ± 15.38	516.9 ± 35.09
Internal Diameter, Systole (mm)	1.85 ± 0.04	1.77 ± 0.05	1.93 ± 0.09	2.08 ± 0.12	1.81 ± 0.11	2.03 ± 0.10	1.93 ± 0.05	4.41 ± 0.15 ****
Internal Diameter, Diastole (mm)	3.48 ± 0.05	3.33 ± 0.05 *	3.44 ± 0.13	3.43 ± 0.11	3.33 ± 0.13	3.46 ± 0.08	3.51 ± 0.06	4.8 ± 0.14 ****
Volume; Systole (mL)	10.41 ± 0.61	9.50 ± 0.70	11.94 ± 1.28	14.68 ± 2.05	10.17 ± 1.48	13.63 ± 1.77	11.62 ± 0.84	88.86 ± 6.87 ****
Volume; Diastole (mL)	50.15 ± 1.61	45.16 ± 1.73	49.43 ± 4.12	49.00 ± 3.83	45.72 ± 4.04	49.70 ± 2.89	51.21 ± 1.95	108.36 ± 7.49 ****
Stroke Volume (mL)	39.4 ± 1.41	35.66 ± 1.14	35.67 ± 2.52	34.32 ± 1.86	34.63 ± 2.73	36.08 ± 1.23	38.36 ± 1.79	19.5 ± 1.52 ****
Ejection Fraction (%)	78.63 ± 1.72	79.14 ± 0.86	73.13 ± 1.84	71.09 ± 2.06	76.75 ± 3.06	73.33 ± 1.96	75.02 ± 2.30	18.23 ± 1.57 ****
Fractional Shortening (%)	46.46 ± 1.43	46.71 ± 0.79	42.05 ± 1.42	39.7 ± 1.64	44.98 ± 2.37	41.65 ± 1.63	43.61 ± 1.85	8.26 ± 0.75 ****
Cardiac Output (mL/min)	22.9 ± 0.94	20.35 ± 0.47 *	21.37 ± 1.40	20.23 ± 1.09	20.98 ± 1.99	20.84 ± 0.58	22.96 ± 1.22	9.89 ± 0.97 ****
LV Mass (mg)	77.45 ± 5.08	62.31 ± 2.62 *	88.24 ± 6.11	79.08 ± 4.65	76.38 ± 6.28	75.94 ± 1.92	88.09 ± 4.80	108.23 ± 6.03 *
LV Mass Cor (mg)	61.96 ± 4.07	49.85 ± 2.09 *	70.59 ± 4.89	63.26 ± 3.72	61.10 ± 5.03	60.75 ± 1.53	70.47 ± 3.84	86.58 ± 4.82 *
LV Anterior Wall Diameter, Systole (mm)	1.26 ± 0.06	1.07 ± 0.02 **	1.40 ± 0.05	1.19 ± 0.03 **	1.33 ± 0.05	1.17 ± 0.02 **	1.35 ± 0.06	0.74 ± 0.02 ****
LV Anterior Wall Diameter, Diastole (mm)	0.81 ± 0.03	0.69 ± 0.02 **	0.90 ± 0.04	0.79 ± 0.02*	0.83 ± 0.04	0.78 ± 0.02	0.85 ± 0.04	0.70 ± 0.03 *
LV Posterior Wall Diameter, Systole (mm)	1.02 ± 0.03	0.92 ± 0.02 *	1.03 ± 0.04	0.92 ± 0.03 *	1.02 ± 0.05	0.90 ± 0.03 *	1.02 ± 0.03	0.57 ± 0.03 ****
LV Posterior Wall Diameter, Diastole (mm)	0.56 ± 0.02	0.56 ± 0.02	0.63 ± 0.02	0.64 ± 0.04	0.60 ± 0.04	0.60 ± 0.03	0.63 ± 0.03	0.48 ± 0.03 **

Comparisons were performed by Two-way ANOVA with Turkey’s post hoc multiple comparisons test. *: comparing Ctrl and *Eprs1* cKO-homo *: *p* < 0.05; **: *p* < 0.01; ****: *p* < 0.0001.

## Data Availability

Data are contained within the article and Appendix A.

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
