# Peer review of "Cardiomyocyte-Specific Loss of Glutamyl-prolyl-tRNA Synthetase Leads to Disturbed Protein Homeostasis and Dilated Cardiomyopathy"

_cells, 2023, doi:10.3390/cells13010035_

Round 1
Reviewer 1 Report
Comments and Suggestions for Authors
The authors of this manuscript investigate the role of EPRS1 in cardiomyocytes. Whereas heterozygous cardiomyocytes specific EPRS1 KO mice do not show any severe cardiac phenotypes either at baseline or after pressure overlaod, homozygous knockout mice develop severe cardiac dysfunction and die within a month after tamoxifen treatment. The manuscript is well written, the data sound and of general interest to the field given the recent development of EPRS1 inhibitors. I have the following comments.
1) It is interesting that a variety of sarcomeric proteins are induced at the mRNA level after deletion of EPRS1 together with several transcription factors (Figure 4A). However, follow up experiments are completly missing. The authors should perform additional experiments to identify the key transcription factor that is responsible for those induction, i.e. luciferase reporter assays in after EPRS1 deletion. This should also provide more insights into the uncoupling of transcription and translation
2) In addition, the authors should test if global translation rate is altered in their knockout models, i.e. by using puromycin incorporation assays or polysome profiles. The induction of both mTOR signaling as well as several ribosomal proteins would suggest massive dysregulation of protein synthesis.
Author Response
Dear Reviewer,
Thank you for your constructive comments and suggestions to improve our research work and current manuscript. Please find our response letter for all four reviewers and responses specifically for your questions below.
Best,
Peng
Reviewer 1:
The authors of this manuscript investigate the role of EPRS1 in cardiomyocytes. Whereas heterozygous cardiomyocytes specific EPRS1 KO mice do not show any severe cardiac phenotypes either at baseline or after pressure overload, homozygous knockout mice develop severe cardiac dysfunction and die within a month after tamoxifen treatment. The manuscript is well written, the data sound and of general interest to the field given the recent development of EPRS1 inhibitors. I have the following comments.
1) It is interesting that a variety of sarcomeric proteins are induced at the mRNA level after deletion of EPRS1 together with several transcription factors (Figure 4A). However, follow up experiments are completely missing. The authors should perform additional experiments to identify the key transcription factor that is responsible for those induction, i.e. luciferase reporter assays in after EPRS1 deletion. This should also provide more insights into the uncoupling of transcription and translation
Response: Thanks for the constructive comments from the reviewer. We observed multiple transcription factors coding mRNAs were significantly increased 2 weeks post Eprs cKO (see Figure 4A), suggesting a potential integrated transcriptional reprogramming that may not be driven by a single key transcription factor. Figure 4A demonstrates that at least 28 transcription factors (TFs) were significantly induced as a compensatory response or secondary effect, including multiple members of KLF TFs (Klf2, Klf4, Klf6), ATF TFs (Atf3, Atf4, Atf5), CEBP TFs (Cebpd, Cebpg), Myc, among others. For instance, ATF4 is a stress-responsive transcription factor induced at both transcription and translation levels by integrated stress response (in this case triggered by amino acid starvation due to EPRS loss of function and insufficient ligation of proline and glutamic acid to their cognate tRNAs) as indicated by activated phosphorylation of eIF2a in Figure 5D. Therefore, ATF4 can be considered as a potential key effector transcription factor in Eprs cKO hearts. As shown in Figure 3A, Slc7a5, Atf5, Mthfd2, Gdf15, Aars, Asns, etc. are well-established ATF4 downstream target genes as readout of integrated stress response pathways. This is consistent with our recent publication in Cardiovascular Research (PMID: 37522353), showing ATF4 is a key effector transcription factor in a mitochondrial protein Fam210a cKO mouse model. We added these points to the discussion.
2) In addition, the authors should test if global translation rate is altered in their knockout models, i.e. by using puromycin incorporation assays or polysome profiles. The induction of both mTOR signaling as well as several ribosomal proteins would suggest massive dysregulation of protein synthesis.
Response: We performed a puromycin incorporation assay in Eprs1 cKO hearts (2 weeks after tamoxifen injection) compared to control hearts and found a mild decrease in global translation at the early stage. We included the data (also figure legend and material & method part) as Figure S3. We did not do the experiment for the late stage (4 weeks after tamoxifen injection) similar to our mass spectrometry analysis to avoid robust secondary effects.

Reviewer 2 Report
Comments and Suggestions for Authors
This paper sets to characterise the phenotypic changes in cardiomyocyte-specific Eprs1 Using a multi-omics approach. A very good paper that shows progress in this field of study.
The introduction is well written and adds the background to the study quite well. More information on the role of Tamoxifen is needed.
Methods:
· Temperature-controlled room – what temperature was it maintained at?
· How was cell size quantified using ImageJ? Was this performed using a pre-designed macro or measuring tools? Would it be possible to add this information on an Appendix?
Results
· The figures are well presented and show consistent data
· N=4 hearts per group is only just enough to collect enough significant data. Is there any reason why the study stopped at 4? Although there is not much data deviation, it would have been good to see more repeats
· Table 2: please add what s and d after volume stand for in the table legend
· Figure 6: The controls for 4 weeks are not presented. Can you present controls for both 2 and 4 weeks?
o What is the second band in alpha-STK38L? No explanation is offered for it in text other than it being non-significant. More explanation needed to justify that this western is not contaminated
The discussion is well written and delineates the study and its main findings well while correlating the discoveries with literature.
Author Response
Dear Reviewer,
Thank you for your constructive comments and suggestions to improve our research work and current manuscript. Please find our response letter for all four reviewers and responses specifically for your questions below.
Best,
Peng
Reviewer 2:
This paper sets to characterise the phenotypic changes in cardiomyocyte-specific Eprs1 Using a multi-omics approach. A very good paper that shows progress in this field of study.
The introduction is well written and adds the background to the study quite well. More information on the role of Tamoxifen is needed.
Methods:
Temperature-controlled room – what temperature was it maintained at?
Response: The range of mouse room temperature is 22 ºC.
How was cell size quantified using ImageJ? Was this performed using a pre-designed macro or measuring tools? Would it be possible to add this information on an Appendix?
Response: After staining with WGA-Alex Fluor-488, the cross-sectional areas were imaged by the upright fluorescence microscope (Olympus), and the scale bar was added to each image. The CM size was quantified using ImageJ software. Briefly, the images are imported into the software and the scale was set based on the scale bar inside each image. A pre-designed macro, Polygon selections, was used to select the area surrounded by the WGA-positive membranes. Finally, the areas inside the WGA-positive membranes were measured as the size of cardiomyocytes. We already updated this part in the method section.
Results
N=4 hearts per group is only just enough to collect enough significant data. Is there any reason why the study stopped at 4? Although there is not much data deviation, it would have been good to see more repeats
Response: We observed robust phenotypes in the Eprs homozygous cKO mouse model and low variability in the collected data with N=4 mice for WGA staining. >300 and >330 CMs were analyzed from the mice for Figure 1E and 2D, respectively. We observed high statistical significance in the large number of CMs from N=4 mice, so we did not include additional mice in the analysis. In most of other figures and related data, we used N>5 mice for other experiments.
Table 2: please add what s and d after volume stand for in the table legend
Response: We have added the full term of the s and d as “Systole” and “Diastole” in the two data tables.
Figure 6: The controls for 4 weeks are not presented. Can you present controls for both 2 and 4 weeks?
Response: We are sorry for the confusion. In most cases of the Eprs homozygous cKO mouse model, we actually used control aMHCMCM mice at 4 weeks post-TMX injection. We have clearly described this point in the figure legends. Figure 2C (previous Figure 2B) contained control aMHCMCM mice for both 2 weeks and 4 weeks and indicated 2 weeks vs. 4 weeks does not show a significant difference for control aMHCMCM mice in terms of ejection fraction. So we only use controls at 4 weeks to reduce the number of mice used as suggested by the University animal usage recommendation.
What is the second band in alpha-STK38L? No explanation is offered for it in text other than it being non-significant. More explanation needed to justify that this western is not contaminated
Response: We are sorry for the confusion. The molecular weight of STK38L is 54 KDa. The upper non-specific band is nearly 70 KDa. “ns” means “non-specific” rather than “non-significant.” (description included in the figure legend).

Reviewer 3 Report
Comments and Suggestions for Authors
This manuscript reported the importance of EPRS1 enzyme in maintaining cardiomyocyte and heart function using a cardiomyocyte-specific Eprs1 inducible knockout mouse model. The manuscript is well written and structured but needs a clarification on the following point:
The authors observed by mass spectrometry analysis a 4-fold of EPRS1 protein decrease after two weeks post knockout (Figure 5C), indicating that the half-life of the protein is ~1 week in vivo. However, the western blot analysis (Figure 5D) shows that after 4 weeks post knockout the EPRS1 protein level increases, or at least, its maintained. Moreover, Figure 1B shows that EPRS1 protein level was reduced by 90% in homozygous Eprs1cKO hearts 2 weeks after knockout (Figure 1B). This is puzzling and difficult to correlate with cardiac function of Eprs1 cKO mice observed throughout the different time points.
Author Response
Dear Reviewer,
Thank you for your constructive comments and suggestions to improve our research work and current manuscript. Please find our response letter for all four reviewers and responses specifically for your questions below.
Best,
Peng
Reviewer 3:
This manuscript reported the importance of EPRS1 enzyme in maintaining cardiomyocyte and heart function using a cardiomyocyte-specific Eprs1 inducible knockout mouse model. The manuscript is well written and structured but needs a clarification on the following point:
The authors observed by mass spectrometry analysis a 4-fold of EPRS1 protein decrease after two weeks post knockout (Figure 5C), indicating that the half-life of the protein is ~1 week in vivo. However, the western blot analysis (Figure 5D) shows that after 4 weeks post knockout the EPRS1 protein level increases, or at least, its maintained. Moreover, Figure 1B shows that EPRS1 protein level was reduced by 90% in homozygous Eprs1cKO hearts 2 weeks after knockout (Figure 1B). This is puzzling and difficult to correlate with cardiac function of Eprs1 cKO mice observed throughout the different time points.
Response: After quantifying the protein intensity of the WB results of EPRS in Figure 5C (quantification data was shown below and added in the manuscript), we did observe that the EPRS protein expression was slightly higher at 4 weeks post-Eprs cKO compared to 2 weeks, although it is not statistically significant with a sample size of N=6 in this case. A major potential reason for this phenomenon is that we used whole heart lysates to perform the WB. At 4 weeks post-cKO, more CMs undergo apoptosis and EPRS protein may derive from other cardiac cell types (such as cardiac myofibroblasts, etc.) more strongly. Our previous published work (PMID: 32611237) showed increased EPRS protein expression in the diseased heart tissues (probably in activated cardiac fibroblasts or myofibroblasts) under cardiac stress conditions. This compensatory increase of EPRS protein in non-cardiomyocyte cells may explain the discrepancy between 2-week and 4-week post-Eprs1 cKO in the whole heart lysates.
The low EPRS protein level in homozygous Eprs cKO hearts at 2 weeks post-cKO in Figure 5D is consistent with the WB for homozygous Eprs cKO hearts in Figure 1B and mass spectrometry data in Figure 5C. The knockout efficiency for the whole heart at 2 weeks is ~70-90% and individual variations were observed among different mice.

Reviewer 4 Report
Comments and Suggestions for Authors
The manuscript entitled: "Cardiomyocyte-specific Loss of Glutamyl-prolyl-tRNA Synthetase Leads to Disturbed Protein Homeostasis and Dilated Cardiomyopathy" by Jiangbin Wu and co-authors describes the phenotypic changes in the hearts of heterozygous and homozygous cardiomyocyte-specific Eprs1-inducible knockout mouse models through a classic approach. In addition, omics analysis was performed to determine transcriptome-wide gene and proteomic expression changes in a longitudinal approach. Furthermore, regarding the relevance of the presented results, the authors report a valuable model of dilated cardiomyopathy. Although it is an interesting and extensive piece of experimental work, some changes should be made to strengthen the results and the article's presentation.
Point 1: Throughout the article, results from different weeks of follow-up are mixed. A flowchart should be added to aid readers to clarify the methodology and different approaches at each stage.
Point 2: Although the authors described a similar one in a previous work, I found it challenging to follow the mouse model. A) Readers of this article should understand easily how the deletion of exon 3 and 4 causes a diminished protein expression of Eprs1. Please add a few sentences to clarify this as an expansion of the brief image in Figure 1. B) Similarly, refer to the articles that describe the genotyping of αMHCMerCreMer. C) In the same way, please add a graph of the changes in weight throughout the follow-up as a supplementary figure. D) Please confirm in the materials section that tamoxifen was used in controls and when the euthanasia was performed in the Ctrl animals. This clarification is relevant in 2 and 4-week comparisons. E) Confirm that both sexes were used in the ulterior analysis.
Point 3: One of the most relevant findings in this article is related to the potential effects of CM-specific loss of Eprs1 as a trigger of dilated cardiomyopathy. However, part of the text is related to the possible role of halofuginone despite this molecule not being used anywhere in this article. These statements are confusing for the reader, as an example in lines 255-257: "To mimic the effect of mild inhibition of EPRS1 activity by halofuginone, we subjected the Eprs1cKO-Het mice to transverse aortic constriction (TAC) surgery and compared the phenotypic outcome in control and Eprs1cKO-Het mice." And in lines 266-267 "the supporting observation that low-dose halofuginone-mediated inhibition of prolyl-tRNA synthetase activity does not cause severe cardiotoxicity in the pressure overload-induced HF model." Therefore, A) The authors should be more descriptive in the presented results, focusing on a heart damage model and enhancing the possible similarity between heterozygous Eprs1 and the use of halofuginone in the discussion. B) Add the survival graph of these animals in both conditions. C) A valuable addition of whole heart images (Ctrl, heterozygous, and homozygous; different times of follow-up) would be helpful to strengthen the results. D) Finally, the authors should consider adjusting the text and Figure 1 as the last one to propose the heterozygous conditional deletion as a model of dilated cardiomyopathy in a long-term approach and/or induced by TAC.
Point 4: Related to Figure 2. A) The survival graph requires color changes to be informative, especially in females. B) In the TUNEL assay, the images display low quality for analysis. If changing the image in this area is impossible, please add higher-quality photographs as a supplementary figure. In this image, the positive marks are localized in alpha-actinin-deployed areas. Are these interstitial or cardiomyocyte areas? How can this be interpreted as Eprs1 deletion is CM-specific? C) Does Picrosirius staining show an interstitial pattern, or is it representative of replacement fibrosis related to increased apoptosis? Add a sentence in the discussion about both interpretations.
Point 5: Related to omics analysis. The authors establish a cutoff: Log2FC or a P value. It is hard to understand why the authors used OR more than an AND. The approach used can increase the number of genes identified as differentially expressed compared to simultaneously using both criteria. However, as this analysis has not been described as merely an exploratory study, the relaxed criteria increase the possibility of false positives, as shown in the western blot STK38L in Figure 6. In the same way, another cutoff was used to determine the downregulated proteins in cKO (FC ≤ -0.5) in the comparison of PP protein analysis. This analysis differs from the previous Log2FC < 0.01 or P < 0.05. A) Therefore, the authors should explain all these differences in the cutoffs.
Point 6: Related to Figure 6. This image raised several doubts. The band patterns do not correlate well with the Ponceau staining in the supplementary figure. Additionally, the last lane of the result of Mylk3 expression does not correlate with the graph of relative expression. Therefore, A) an individual immunoblot with three representative samples of each treatment should be included to correlate Ctrl, 2, and 4 weeks accurately. I suggest that the actual image should be moved to a supplementary figure. B) Finally, the authors should indicate whether Alpha or beta-actin was used as a gel loading control.
Point 7: Despite the authors describing a previous report about the polysome interaction of mRNA of fatty acids, it would be valuable to add a sentence (one or two lines) about the changes in metabolism related to heart failure, as expressed in articles such as Lopaschuk GD, Karwi QG, Tian R, Wende AR, Abel ED. Cardiac Energy Metabolism in Heart Failure. Circ Res. 2021 May 14;128(10):1487-1513.
Point 8: Despite the authors considering that the drop in cardiac variables at 2 and 4 weeks is more related to half-lives of mRNA and Eprs1, a paragraph should be added to suggest the possibility of a homeostatic response before the appearance of heart failure signs, as suggested by the increase in Nppa and Nppb at 2 and 4 weeks displayed in omics analysis.
Point 9: The discussion should be enhanced and reoriented. Furthermore, aside from describing the missense variants of EPRS1 in humans, the article presents several results that should be discussed thoroughly. For example, the possibility of senescence induction in cardiac pathological remodeling with specific compensatory responses at the mRNA, as suggested by Cdkn1a, mTOR, and p70S6K results. The authors should reconsider strengthening the discussion based on the presented observations.
Point 10: The authors used the Tukey post-hoc test throughout the statistical analysis when appropriate. However, due to the graphic presentation, the data from 2 and 4 weeks are exclusively compared against the control (ctrl), resembling the approach of a Dunnett test. Are there any additional statistical differences between the two and fourth-week data that may not be depicted in the graphs or images?
Point 11: Revise if genes and proteins are adequately written; italics should be used for genes and RNA. The authors sometimes use capital letters when describing gene/protein letters; this is most common for human nomenclature.
Comments on the Quality of English LanguageCareful grammar and spell check are required to minimize typos: Indentations were added in figure captions, and the Greek letters were missed in lines 336 and 493.
Author Response
Dear Reviewer,
Thank you for your constructive comments and suggestions to improve our research work and current manuscript. Please find our response letter for all four reviewers and responses specifically for your questions below.
Best,
Peng
Reviewer 4:
The manuscript entitled: "Cardiomyocyte-specific Loss of Glutamyl-prolyl-tRNA Synthetase Leads to Disturbed Protein Homeostasis and Dilated Cardiomyopathy" by Jiangbin Wu and co-authors describes the phenotypic changes in the hearts of heterozygous and homozygous cardiomyocyte-specific Eprs1-inducible knockout mouse models through a classic approach. In addition, omics analysis was performed to determine transcriptome-wide gene and proteomic expression changes in a longitudinal approach. Furthermore, regarding the relevance of the presented results, the authors report a valuable model of dilated cardiomyopathy. Although it is an interesting and extensive piece of experimental work, some changes should be made to strengthen the results and the article's presentation.
Point 1: Throughout the article, results from different weeks of follow-up are mixed. A flowchart should be added to aid readers to clarify the methodology and different approaches at each stage.
Response: Thanks for providing detailed comments on the eleven points for us to improve the manuscript. Schematic models were added in Figure 1, 2 to facilitate understanding of the work flow. Figures 3-6 are follow-up transcriptomic and proteomic analysis of homozygous Eprs cKO mice vs. control mice at 2 weeks post tamoxifen injection. We did not use 3 or 4 weeks samples to avoid extensive secondary effects from the late stage HF (this point was added in the result section).
Point 2: Although the authors described a similar one in a previous work, I found it challenging to follow the mouse model. A) Readers of this article should understand easily how the deletion of exon 3 and 4 causes a diminished protein expression of Eprs1. Please add a few sentences to clarify this as an expansion of the brief image in Figure 1. B) Similarly, refer to the articles that describe the genotyping of αMHCMerCreMer. C) In the same way, please add a graph of the changes in weight throughout the follow-up as a supplementary figure. D) Please confirm in the materials section that tamoxifen was used in controls and when the euthanasia was performed in the Ctrl animals. This clarification is relevant in 2 and 4-week comparisons. E) Confirm that both sexes were used in the ulterior analysis.
Response: A) We added a few sentences to clarify the process of generating the Eprs1 cKO mouse model as a detailed explanation of Figure 1A. B) We cited the original article of Circ. Res. that describes the genotyping of αMHCMerCreMer. C) Unfortunately, we did not collect the data of the changes in weight throughout the follow-up. D) We confirmed in the materials section that tamoxifen was used in controls and described the time point of the euthanasia/harvest was performed in the Ctrl animals in figure legends. This clarification is relevant in 2 and 4-week comparisons. Figure 2C (previous Figure 2B) contained control aMHCMCM mice for both 2 weeks and 4 weeks and indicated 2 weeks vs. 4 weeks does not show a significant difference for control aMHCMCM mice in terms of ejection fraction. Other figures (in Figures 2D, E, F, G) used the control aMHCMCM mice with 4 weeks post-TMX injection only to reduce the number of mice used as generally suggested by the University animal usage recommendation. E) We confirm that both sexes were used in the analysis in Figure 2. The number of mice with different sex was described in the figure legends.
Point 3: One of the most relevant findings in this article is related to the potential effects of CM-specific loss of Eprs1 as a trigger of dilated cardiomyopathy. However, part of the text is related to the possible role of halofuginone despite this molecule not being used anywhere in this article. These statements are confusing for the reader, as an example in lines 255-257: "To mimic the effect of mild inhibition of EPRS1 activity by halofuginone, we subjected the Eprs1cKO-Het mice to transverse aortic constriction (TAC) surgery and compared the phenotypic outcome in control and Eprs1cKO-Het mice." And in lines 266-267 "the supporting observation that low-dose halofuginone-mediated inhibition of prolyl-tRNA synthetase activity does not cause severe cardiotoxicity in the pressure overload-induced HF model." Therefore, A) The authors should be more descriptive in the presented results, focusing on a heart damage model and enhancing the possible similarity between heterozygous Eprs1 and the use of halofuginone in the discussion. B) Add the survival graph of these animals in both conditions. C) A valuable addition of whole heart images (Ctrl, heterozygous, and homozygous; different times of follow-up) would be helpful to strengthen the results. D) Finally, the authors should consider adjusting the text and Figure 1 as the last one to propose the heterozygous conditional deletion as a model of dilated cardiomyopathy in a long-term approach and/or induced by TAC.
Response: We appreciate the critical insights from the reviewer. In 2017, an article published in JAHA showed that halofuginone protected hearts from the cardiac stress from pressure overload (TAC surgery), neurohormonal stimulation (angiotensin II/phenylephrine), and acute ischemia-reperfusion injury by activating amino acid starvation pathway (JAHA 2017;6:e004453). We acknowledge that low-dose halofuginone-mediated inhibition of prolyl-tRNA synthetase activity cannot fully recapitulated by heterozygous Eprs cKO. In the latter case, a reduced level of EPRS enzyme still maintains full catalytic activity in ligating proline to corresponding tRNAPro. Also, the CM-specific heterozygous cKO of one allele of Eprs gene cannot recapitulate the cardioprotective effects of halofuginone in reducing pro-fibrotic extracellular matrix protein expression in cardiac fibroblasts. Therefore, we modified the halofuginone-related introduction in the result section and further covered this point in the discussion section.
We did not perform any animal studies using halofuginone because GSK company did these experiments using three HF mouse models, as mentioned above. Therefore, we cannot provide additional data as requested from A-C. We cited the GSK company’s JAHA article in our manuscript. We did not move Figure 1 to the last figure as we don’t consider the heterozygous Eprs cKO mouse model as a potential dilated cardiomyopathy model in the long term because our unpublished observation of heterozygous Eprs global KO mouse model did not develop any heart disease symptoms. We think that the het cKO model is a negative control compared to homo cKO model in triggering spontaneous DCM and HF, which is logically smooth to be integrated together in Figure 1. We appreciate the reviewer’s thoughtfulness.
Point 4: Related to Figure 2. A) The survival graph requires color changes to be informative, especially in females. B) In the TUNEL assay, the images display low quality for analysis. If changing the image in this area is impossible, please add higher-quality photographs as a supplementary figure. In this image, the positive marks are localized in alpha-actinin-deployed areas. Are these interstitial or cardiomyocyte areas? How can this be interpreted as Eprs1 deletion is CM-specific? C) Does Picrosirius staining show an interstitial pattern, or is it representative of replacement fibrosis related to increased apoptosis? Add a sentence in the discussion about both interpretations.
Response: Thanks for these constructive suggestions. A) We changed the color of the graph in Fig 2A to make the data more informative. B) We added one image with the corresponding areas zoomed in to provide more detailed information. From the corresponding areas zoomed in of the TUNEL staining assay, we observed that some TUNEL signals colocalized with a-actinin and some signals localized at a-actinin deployed areas. We think the signals at a-actinin deployed areas came from the dead CMs with degraded a-actinin and some DNA fragments remaining in these areas. Thus, these are still CM areas. The TUNEL and a-actinin co-localized signals came from CMs with early-stage apoptosis. C) We added this point to the discussion as follows: “On the other hand, homozygous Eprs1 cKO leads to compromised global translational arrest at the late stage, as indicated by phosphorylation of eIF2a and subsequent activation of integrated stress response. As a result, CMs underwent apoptosis and cell death while cardiac fibroblasts were activated, leading to replacement fibrosis triggered by CM death and possibly interstitial fibrosis during cardiac pathological remodeling.”
Point 5: Related to omics analysis. The authors establish a cutoff: Log2FC or a P value. It is hard to understand why the authors used OR more than an AND. The approach used can increase the number of genes identified as differentially expressed compared to simultaneously using both criteria. However, as this analysis has not been described as merely an exploratory study, the relaxed criteria increase the possibility of false positives, as shown in the western blot STK38L in Figure 6. In the same way, another cutoff was used to determine the downregulated proteins in cKO (FC ≤ -0.5) in the comparison of PP protein analysis. This analysis differs from the previous Log2FC < -1 (or >1) or P < 0.05. A) Therefore, the authors should explain all these differences in the cutoffs.
Response: We agree with the reviewers’ comments and admit the cut-off is arbitrary based on the nature of the screen in Figure 5 and 6. For both Figure 5 and 6, we can always make the cut-off more stringent or flexible to decrease or increase the number of potential candidates for WB validation as we need to revisit the data.
In Figure 5A, the proteomic mass spectrometry was performed at 2 weeks post tamoxifen injection. We avoided the late stage because of the extensive secondary effects from HF despite many statistically significant and dramatic changes happening. At the early stage, proteomic changes were moderate as no cardiac dysfunction was detected by echocardiography. If we set the overlapped hits with the standard of fulfilling both | Log2FC (cKO/Ctrl) | > 1 and P < 0.05, we will lose many biologically relevant changes (false negative). For example, well known integrated stress response triggered by amino acid starvation (in our case Eprs homo cKO) activates ATF4 transcription factor expression at both transcription and translation levels, leading to increased transcription of the majority of aminoacyl-tRNA synthetases to different extent. In Figure 5C, if we set up | both Log2FC (cKO/Ctrl) | > 1 and P < 0.05, we will only get Eprs1, Iars1, and Mars1 and lost Dars1, Qars1, Rars1, Lars1, Kars1, whose proteins are subject to proteasomal degradation as they reside in the multi-synthetase complex with EPRS1 (without EPRS1 the complex cannot form and component proteins are destabilized). Therefore, we set the cutoff as | Log2FC (cKO/Ctrl) | > 1 or P < 0.05 to get a biologically meaningful full picture with the cost of some false positive. This trade-off is worth better understanding the biological impact of loss-of-function of Eprs1. In the case of | Log2FC (cKO/Ctrl) | > 1, we looked at the dramatically dysregulated proteins. In the case of P < 0.05, we focused on statistically significantly changed proteins though some changes are modest (e.g., | Log2FC | < 0.25). For example, in Figure 5E, many fatty acid and branched-chain amino acid metabolic enzymes were significantly but moderately or modestly downregulated. We would miss these important changes if we set up | both Log2FC (cKO/Ctrl) | > 1 and P < 0.05. Importantly, as the reviewer pointed out, we can perform WB to rule out false negative hits such as STK38L.
We could use Log2FC < -1 or P < 0.05 instead of Log2FC < -0.5, but we would get a very small number of overlapped hits for further validation. As Figure 6A is a screen, we hope to expand potential candidate hits for validation in Figure 6B. Mass spectrometry analysis is also a proteomic screen with limitations of sensitivity and accuracy; moderate changes do not necessarily suggest they are biologically meaningless. Multiple buffering or compensatory mechanisms can counteract primary interventions within a biological system, requiring us to be cautious in ruling out moderate changes by arbitrary mathematical cut-offs. We performed a similar analysis as shown in new Figure S3A using a cut-off of Log2FC<-1 or Log2FC<-1 & P<0.05 and included the data. Our tested 4 proteins are still in the overlapped hits, including TGM1, AOX1, MYLK3, and STK38L.
Point 6: Related to Figure 6. This image raised several doubts. The band patterns do not correlate well with the Ponceau staining in the supplementary figure. Additionally, the last lane of the result of Mylk3 expression does not correlate with the graph of relative expression. Therefore, A) an individual immunoblot with three representative samples of each treatment should be included to correlate Ctrl, 2, and 4 weeks accurately. I suggest that the actual image should be moved to a supplementary figure. B) Finally, the authors should indicate whether Alpha or beta-actin was used as a gel loading control.
Response: The band patterns in TGM1 WB correlate with the Ponceau staining in Figure S2B. We did not show all the Ponceau staining membranes, and we loaded the same amount of total proteins for WB of other proteins. We double-checked the quantification of MYLK3 WB and confirmed graph data was consistent with the initial Image J analysis.
- A) We have run low in the original tissue lysate samples and cannot rerun the representative samples as suggested, unfortunately. For Figure 6B, we ran 4-week samples and control plus 2-week samples in two separate gels following the same procedure (voltage, running time, antibody incubation, exposure time, etc.). We think they are comparable after normalization. Beta-actin was used as a gel loading control. Alpha means anti-beta-actin. We are sorry for the confusion.
Point 7: Despite the authors describing a previous report about the polysome interaction of mRNA of fatty acids, it would be valuable to add a sentence (one or two lines) about the changes in metabolism related to heart failure, as expressed in articles such as Lopaschuk GD, Karwi QG, Tian R, Wende AR, Abel ED. Cardiac Energy Metabolism in Heart Failure. Circ Res. 2021 May 14;128(10):1487-1513.
Response: We cited the suggested paper and added a statement as follows: On the other hand, we cannot rule out the possibility of a secondary effect downstream of HF caused by loss-of-function of Eprs1 as fatty acid oxidation is often compromised and related metabolic enzyme expression is reduced in hypertensive cardiomyopathy and ischemic heart failure.
Point 8: Despite the authors considering that the drop in cardiac variables at 2 and 4 weeks is more related to half-lives of mRNA and Eprs1, a paragraph should be added to suggest the possibility of a homeostatic response before the appearance of heart failure signs, as suggested by the increase in Nppa and Nppb at 2 and 4 weeks displayed in omics analysis.
Response: We added the following point in the discussion: Alternatively, a homeostatic response may maintain cardiac function before the appearance of heart failure symptoms, as suggested by the significant increase in Nppb, Nppa, and Myh7 at 2 weeks as displayed in RNA-seq (Figure 3A) or mass spectrometry (Figure 5A) analysis.
Point 9: The discussion should be enhanced and reoriented. Furthermore, aside from describing the missense variants of EPRS1 in humans, the article presents several results that should be discussed thoroughly. For example, the possibility of senescence induction in cardiac pathological remodeling with specific compensatory responses at the mRNA, as suggested by Cdkn1a, mTOR, and p70S6K results. The authors should reconsider strengthening the discussion based on the presented observations.
Response: We added a paragraph in the discussion as follows: Additionally, we cannot rule out the possibility of senescence induction in cardiac pathological remodeling with specific compensatory responses at the mRNA level, as indicated by increased expression of Cdkn1a as a key p53 pathway effector 24 at the early stage (Figure 4A, B). mTORC1 pathway was activated at the late stage but not at the early stage, suggesting that Eprs1 cKO CMs were quiescent at the early stage with p53 activation and mTOR inhibition and became senescent at the late stage with both p53 and mTOR activation 24.
Point 10: The authors used the Tukey post-hoc test throughout the statistical analysis when appropriate. However, due to the graphic presentation, the data from 2 and 4 weeks are exclusively compared against the control (ctrl), resembling the approach of a Dunnett test. Are there any additional statistical differences between the two and fourth-week data that may not be depicted in the graphs or images?
Response: We have added the statistical data (P value) for comparing 2 weeks vs 4 weeks in Figures 2C, 2D, 2E, 4B, 4C, 5B, 5D, 5F, and 5G. Please find the updated statistical data in these figures.
Point 11: Revise if genes and proteins are adequately written; italics should be used for genes and RNA. The authors sometimes use capital letters when describing gene/protein letters; this is most common for human nomenclature.
Response: Thanks for the suggestion. We corrected the nomenclature following the comment and general rule for the mouse: Gene symbols are italicized, with only the first letter in upper-case (e.g., Eprs1). Protein symbols are not italicized, and all letters are in upper-case (e.g., EPRS1).

Round 2
Reviewer 1 Report
Comments and Suggestions for Authors
no futher comments
Reviewer 3 Report
Comments and Suggestions for Authors
The authors have satisfactorily addressed my previous questions and did as well improved the clarity of the manuscript.
Comments on the Quality of English LanguageSome of the sentences in the text could benefit from a better english editting.
Reviewer 4 Report
Comments and Suggestions for Authors
The authors have satisfactorily addressed the raised comments and questions, significantly enhancing the understanding of the initial limitations and inquiries in the manuscript.
Comments on the Quality of English LanguageTwo suggestions to consider are:
a) Ensuring consistency in the writing style—occasionally, it switches between first-person and third-person perspectives.
b) Verifying that the additions made after the review match the paragraph style employed throughout the manuscript.